# ComGAN: Unsupervised Disentanglement and Segmentation via Image Composition

**Rui Ding**
Central South University
ruiding@csu.edu.cn

**Kehua Guo**[*]
Central South University
guokehua@csu.edu.cn

**Xiangyuan Zhu**
Central South University
zhuxiangyuan@csu.edu.cn

**Zheng Wu**
Central South University
wuzhenghuse@gmail.com

**Liwei Wang**
Central South University
wang.liwei@csu.edu.cn

## Abstract

We propose ComGAN, a simple unsupervised generative model, which simultaneously generates realistic images and high semantic masks under an adversarial loss and a binary regularization. In this paper, we first investigate two kinds of trivial solutions in the compositional generation process, and demonstrate their source is vanishing gradients on the mask. Then, we solve trivial solutions from the perspective of architecture. Furthermore, we redesign two fully unsupervised modules based on ComGAN (DS-ComGAN), where the disentanglement module associates the foreground, background and mask with three independent variables, and the segmentation module learns object segmentation. Experimental results show that (i) ComGAN's network architecture effectively avoids trivial solutions without any supervised information and regularization; (ii) DS-ComGAN achieves remarkable results and outperforms existing semi-supervised and weakly supervised methods by a large margin in both the image disentanglement and unsupervised segmentation tasks. It implies that the redesign of ComGAN is a possible direction for future unsupervised work.[1]

## 1 Introduction

Generative adversarial networks (GANs) [1] have been successful in realistic image generation [2, 3], and recent work on GAN-based image composition [4, 5, 6, 7, 8, 9] has shown that GANs identify and disentangle a class of objects from the background. To direct that GANs learn the distinction between foreground and background, this kind of work commonly requires the following assumption:

**Assumption 1 (Image composition [10] )** *An image $x$ taken from the world is typically composed of foreground $x_f$ and background $x_b$, which can be decomposed by the following equation:*

$$x = x_f \odot x_m + x_b \odot (1 - x_m),  \tag{1}$$

*where $x_m$ is the mask, and the $\odot$ denotes element wise multiplication operator.*

It is notable that the Assumption 1 is mild and achieves significant outcomes in many areas. For example, the models [11, 12] capture meaningful foreground segmentation masks via exactly coupling different foregrounds and backgrounds. C3-GAN [13] proposes a scene decomposition-based method to enforce the model to learn the features of foreground objects. Yang et al. [14, 15] detect moving

---

[*]Corresponding author

[1]Code and data are available at https://github.com/Ruiding1/ComGAN

objects in video by minimizing the mutual information of foreground-background and training in an adversarial manner. A class of methods [16, 17] explores and perturbs the latent space of pre-trained GANs to find foreground masks. However, it is observed that under the Assumption 1, the generation process is always accompanied by trivial solutions [10, 18, 11, 19, 20, 21, 22, 12]. Trivial solutions [11] can be considered as meaningless masks generated by models. To our knowledge, existing works do not indicate that the source of the trivial solutions and they alleviate this issue in two ways. One way is to add supervised information [23, 21, 19, 18], such as CGN [19] avoids trivial solutions by adding pre-trained U2-Net [24]. Another is to design clever regularization and fine-tune the parameters [10, 11, 20, 22, 12]. For example, PerturbGAN [11] proposes a regularization based on image recombination with arbitrarily small relative shifts. Yang et al. [20] propose a regularization based on mutual information maximization. These methods are sensitive to object scale, object category and datasets, and therefore consume computational resources and time costs to perform hyperparameters fine-tuning. For our solution, we propose an unsupervised generative model called ComGAN, which simultaneously generates realistic images and high semantic masks. In addition, our model effectively avoids trivial solutions from the perspective of architecture, which implies that the model does not require any supervised information and explicit regularization. More specifically, ComGAN is a generic way to generalize two typical image compositional generation methods and alleviate the shortcomings coming from the above two methods. Furthermore, we place unique restrictions on each module, which help the model avoid trivial solutions.

Trivial solutions also have a negative role in related tasks. For example, in the image disentanglement, low semantic masks cause foreground and background conflicts [25, 26], which reduce the synthesis quality. In unsupervised segmentation, the regularization to alleviate trivial solutions may induce overfitting of the segmentation masks [14, 15, 12]. To highlight the flexibility and robustness of ComGAN, we extend it to the above two tasks. The ComGAN-based variant is called DS-ComGAN, which contains both disentanglement and segmentation modules. For multi-factor disentanglement, existing methods [26, 21, 27] rely on additional supervised information to learn the distinction of image regions. Notice that ComGAN achieves foreground-background disentanglement in an unsupervised way. As a result, the disentanglement module is designed as follows: we add more global information to the shared features in ComGAN and maximize the mutual information between variables and images. This module simplifies the previous hierarchical generative network and outperforms the state-of-the-art (SOTA) semi-supervised and weakly supervised image disentanglement methods. For unsupervised segmentation, existing methods [14, 15, 11, 12] rely on strong assumptions, such as that the foreground and background are largely independent, which limits their applicability. Different from these methods, we train a segmentation network using the images and semantic masks synthesized by the disentanglement module. By adversarial training strategy, the image distribution and the mask distribution are aligned. Furthermore, a consistency regularization is introduced to ensure that the predicted masks are consistent with the inputs. This segmentation module relies only on mild Assumption 1 and outperforms the SOTA unsupervised segmentation methods.

The main contributions are highlighted as follows:

1. To the best of our knowledge, we are the first to find that the source of trivial solutions is vanishing gradients on the mask. Furthermore, we propose ComGAN, a simple unsupervised generative model, which is the first to solve trivial solutions from the perspective of architecture and achieve foreground-background disentanglement with only an adversarial loss and a binary regularization.

2. We propose DS-ComGAN, a variant network based on ComGAN, which achieves image disentanglement and object segmentation in a fully unsupervised manner. Experiments show that DS-ComGAN is robust to various datasets and outperforms SOTAs in both tasks.

## 2   Related Work

**GAN-based Image Composition.** Image composition can be regarded as combining multiple visual areas to construct a realistic image [28]. A series of typical methods built on GANs [1] perform image composition by utilizing Assumption 1 to achieve various functions. LR-GAN [10] composites realistic images by learning to generate the backgrounds and foregrounds separately and recursively. FineGAN [25] hierarchically composites images, which disentangles the background, object shape, and object appearance. PerturbGAN [11] learns segmentation masks by training a generative model of a layered scene and composing foregrounds and backgrounds. Similarly, SEIGAN [18] performs

the cut, paste and inpaint operations consistent with semantic information to obtain segmentation masks. Although the compositional image generation performs excellently in several tasks under Assumption 1, these methods require additional regularization and hyperparameters tuning to avoid trivial solutions.

**Controllable Image Disentanglement.** The goal of controllable image disentanglement is to disentangle factors of variation (e.g., object shape and object appearance) towards controllable generation. InfoGAN [29] proposes semantic image generation control by imposing regularization of mutual information. A series of works [25, 30, 31, 26] hierarchically generates images and disentangles the background, the object's shape and its appearance by bounding box annotation. MixNMatch [31] extends the FineGAN [25], which disentangles and encodes four variable factors (background, object pose, shape, and texture) from real images. Along this direction, PartGAN [30] achieves part-level decomposition by learning a part generator. Benny and Wolf [21] further propose a complex GAN-based generative model, which requires a set of clean background images to simultaneously solve several tasks including image disentanglement and unsupervised segmentation. These methods [32, 33] learn the independent latent characteristics of an object, especially its appearance and pose. CGN [19] decomposes the image generation process into independent causal mechanisms, and allows for generating counterfactual images. SSC-GAN [27] is a semi-supervised single-stage generative model considers three factors of variation via class labels, namely independent variables, cross-class variables, and class variables. Although these methods exhibit strong performance, they all rely on additional supervised information to achieve multi-factor disentanglement.

**Unsupervised Object Segmentation.** Unsupervised object segmentation aims to extract a useful interpretation from an image in an unsupervised way. W-net [34] extracts foreground objects through an encoder-decoder framework and minimizes reconstruction errors. ReDO [35] captures the mask of the object, relying on the assumption that changing the foreground objects does not affect the overall data distribution. Assuming that the foreground and background are largely independent, these methods [14, 15, 11, 12] obtain segmentation masks by reconstructing the realistic images, that is, precisely combining the different foregrounds and backgrounds. Labels4Free [12] performs unsupervised object segmentation based on the idea that exact coupling between foreground and background is highly non-trivial. A similar assumption is implemented In method [36], where IEM learns image segmentation from an information information-theoretic by maximizing inpainting error. Similarly, Yang et al. [20] synthesize paired realistic images and segmentation masks by maximizing mutual information between generated images and latent variables. These methods frequently require stringent assumptions and are not necessarily applicable to complex real-world data.

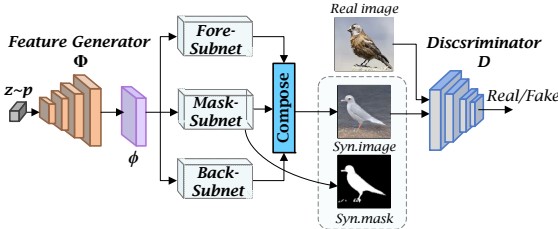

Figure 1: **Overview of ComGAN.** The generator of ComGAN is composed of the feature generator $\Phi$ and three independent subnetworks. Three subnetworks synthesize the foreground, background and mask with features $\phi$ as a shared input. The 'compose' indicates the utilization of Assumption 1 to compose the image. The discriminator $D$ distinguishes real images from fake ones.

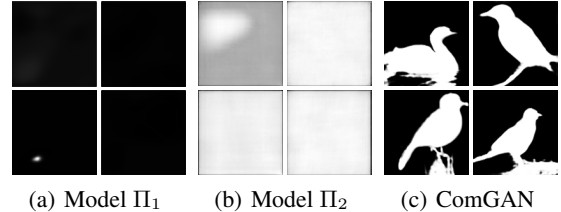

(a) Model $\Pi_1$  (b) Model $\Pi_2$  (c) ComGAN

Figure 2: **Comparison results of the synthesized mask.** ComGAN synthesizes high semantic masks, while the masks synthesized by models $\Pi_1$ and $\Pi_2$ both converge to trivial solutions. Note that $\Pi_1$ has the same structure as [25, 30, 31, 26], and $\Pi_2$ has a similar structure to [11, 19]. All models include identical adversarial loss and binary regularization $\beta L_{binary}$ from [19]. For each model: (left) $\beta = 0.1$, (right) $\beta = 1$.

## 3 Proposed Approach

### 3.1 Trivial Solutions and ComGAN

**Source of trivial solutions.** Two kinds of trivial solutions were introduced in [11]. The first trivial solution degrades the whole scene into foreground or background, and the mask is always full or empty. The second trivial solution makes the foreground and background identical, and the mask is meaningless for the composite scene. Then, we state the source of two kinds of trivial solutions.

**Lemma 1** *(**Vanishing gradients on the mask**). Let $L_{all}$ be the overall loss and $\bar{x}_m$ be the synthesized mask. Consider a model that composes images utilizing Assumption 1. There exist vanishing gradients on the mask, i.e. $\partial L_{all} / \partial \bar{x}_m = 0$ if and only if the model converges to two kinds of trivial solutions.*

*Proof:* See Appendix B.1. $\qquad\square$

**Network architecture of ComGAN.** An overview of our proposed ComGAN is illustrated in Fig. 1. The generator of ComGAN consists of two parts. The feature generator denoted as $\Phi$ synthesizes features conditioned on the variable $z$, i.e. $\phi = \Phi(z)$ where $z$ is sampled from a prior distribution $p_0$. The three subnetworks are defined as $\mathcal{F}$, $\mathcal{B}$ and $\mathcal{M}$. They synthesize the foreground, background and mask, that is, $\bar{x}_f = \mathcal{F}(\phi)$, $\bar{x}_b = \mathcal{B}(\phi)$ and $\bar{x}_m = \mathcal{M}(\phi)$. Driven by Assumption 1, the composite image $\bar{x}$ is defined as:

$$\bar{x} = \mathcal{F}(\Phi(z)) \odot \mathcal{M}(\Phi(z)) + \mathcal{B}(\Phi(z)) \odot (1 - \mathcal{M}(\Phi(z))) \qquad (2)$$

**Remark 1** *This form generalizes two typical image compositional generation methods:*

• *If only $\Phi(\cdot)$ in $\mathcal{B}$ is an identity map, i.e. $\mathcal{B}(\Phi(z)) = \mathcal{B}(z)$, then this form is equivalent to the model $\Pi_1$, that is, two independent generators synthesize a composite image where shared features exist in the foreground and mask generation. For example, FineGAN [25], C3-GAN [13] and Labels4Free [12], etc. can be formulated as the model $\Pi_1$.*

• *If $\Phi(\cdot)$ is an identity map, i.e. $\Phi(z) = z$, then this form is equivalent to the model $\Pi_2$, that is, three independent generators synthesize a composite image where foreground, background and mask are generated by three generators respectively. For example, PerturbGAN [11] and CGN [19], etc. can be formulated as the model $\Pi_2$.*

From the above observations, the two typical methods have a common shortcoming, that is, models $\Pi_1$ and $\Pi_2$ both contain an independent background generation process. The term ignores the association of foreground, background and mask on the features. Moreover, if the model converges to trivial solutions, then the model degrades to an original GAN that suffers from known shortcomings, such as mode collapse. As a result, our proposed method is a generic way and mitigates the above shortcoming. Then, the adversarial training loss $L_D^{adv}$ is defined as follows:

$$\min_{\Phi, \mathcal{F}, \mathcal{B}, \mathcal{M}} \max_{D} \mathcal{L}_D^{adv} = \mathbb{E}_{x \sim p_{\text{data}}} \left[ \log \left( D(x) \right) \right] + \mathbb{E}_{z \sim p_0} \left[ 1 - \log D(\bar{x}) \right]. \qquad (3)$$

Obviously, Lemma 1 points us to the limitation of the compositional generation under Assumption 1. Then, we illustrate how to address the issue from the perspective of architecture.

**Theorem 1** *Given a generation $G_\theta$ composed of a decoder $\Phi_{\theta_\phi} : \mathcal{Z} \to \phi$ and three subnets $\mathcal{F}_{\theta_f}$, $\mathcal{B}_{\theta_b}$ and $\mathcal{M}_{\theta_m} : \phi \to \mathcal{X}$. Let $D$ be a discriminator and $D^*(G^*(\cdot))$ be Nash equilibrium. If the composite images satisfy $\bar{x} = \mathcal{F}(\Phi(z)) \odot \mathcal{M}(\Phi(z)) + \mathcal{B}(\Phi(z)) \odot (1 - \mathcal{M}(\Phi(z)))$, $\|D(G(\cdot)) - D^*(G^*(\cdot))\| < \epsilon$, $\max\{\mathbb{E}_{\phi \sim p(\Phi(z))} \left[ \|J_{\theta_f} \mathcal{F}(\phi)\|_2^2 \right], \mathbb{E}_{\phi \sim p(\Phi(z))} \left[ \|J_{\theta_b} \mathcal{B}(\phi)\|_2^2 \right]\} \leq \delta^2$ and $\|\nabla_\theta L_D^{adv}\|_2 \geq \sigma$, then*

$$\left\| \nabla_{(\theta_\phi, \theta_m)} \mathbb{E}_{z \sim p(z)} \left[ \log \left( 1 - D \left( \mathcal{F}(\Phi(z)) \right) \right) \right] \right\|_2^2 > \sigma^2 - \frac{\delta^2 \epsilon^2}{(1/2 - \epsilon)^2}. \qquad (4)$$

*Proof:* See Appendix B.1. $\qquad\square$

We denote that $\rho = \sigma^2 - \frac{\delta^2 \epsilon^2}{(1/2-\epsilon)^2}$ and trivial masks as $\bar{x}_m^*$. If $\rho > 0$, then the updated masks are $\bar{x}_m^+ = \mathcal{M}_{\theta_m^+}(\Phi_{\theta_\phi^+}(z))$, which means the model escape from the first trivial solution, i.e. $\bar{x}_m^+ \neq \bar{x}_m^*$.

**Corollary 1** *The following modules restrictions help the model avoid trivial solutions: the $\mathcal{F}$ and $\mathcal{B}$ are lightweight and differential, the $\mathcal{M}$ is a shallow network and the capacity of $\Phi$ is enough.*

*Proof:* See Appendix B.1. $\qquad\square$

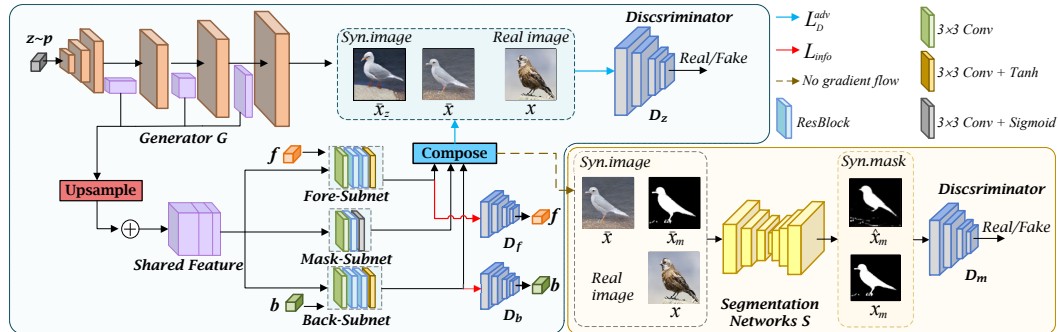

Figure 3: **Overview of DS-ComGAN.** DS-ComGAN consists of two modules: (i) the disentanglement module (blue); (ii) the segmentation model (yellow). The feature generator $G$ synthesizes $\bar{x}_z$ to perform adversarial training with the discriminator $D_z$ to contain more global features. Shared features are extracted from $G$ and fed to three subnetworks $\mathcal{F}$, $\mathcal{B}$ and $\mathcal{M}$. Driven by Assumption 1, foreground, background, and mask are synthesized and composited into an image $\bar{x}$. The module implements image disentanglement by the adversarial loss $L_D^{adv}$ and the mutual information regularization $L_{info}$. Since the model synthesize high semantic masks, the segmentation module performs unsupervised segmentation with image distribution alignment and mask distribution alignment via adversarial training.

## 3.2 The Disentanglement Module of DS-ComGAN

Image generation is controlled by the mask variable $z$, foreground variable $f$ and background variable $b$, where $f$ and $b$ are both $N$ dimensional random one-hot vectors. The generator consists of two parts, namely a feature generator denoted as $G$ and three subnetworks defined as $\mathcal{F}$, $\mathcal{B}$ and $\mathcal{M}$.

**Global information for shared features.** The features should contain global information so that the model captures a wide range of distinctions between foreground and background, rather than focusing on abrupt regions. To add global information to the shared features of $G$, we synthesize realistic images for adversarial training, that is, $\bar{x}_z$ is synthesized by $G(z)$ and fed to the discriminator $D_z$. Then, we extract the activation map from the last three layers inside $G$, denoted as $a_1$, $a_2$ and $a_3$. The shared features with global information can be written as follows:

$$\phi = \mathbb{U}(a_1) \oplus \mathbb{U}(a_2) \oplus \mathbb{U}(a_3), \tag{5}$$

where $\mathbb{U}$ is a spatially upsample to maintain the same size for all activation maps and $\oplus$ is a concatenation along the channel dimension. In order to control image disentanglement, $\mathcal{F}$, $\mathcal{B}$ and $\mathcal{M}$ synthesize foreground, background, and mask conditioned on $f$, $b$ and $z$, that is, $\bar{x}_f = \mathcal{F}(\phi, f)$, $\bar{x}_b = \mathcal{B}(\phi, b)$ and $\bar{x}_m = \mathcal{M}(\phi)$. Driven by Assumption 1, the composite image $\bar{x}$ are defined as:

$$\bar{x} = \mathcal{F}(\phi, f) \odot \mathcal{M}(\phi) + \mathcal{B}(\phi, b) \odot (1 - \mathcal{M}(\phi)). \tag{6}$$

Then, the discriminator $D_z$ is trained to classify images as real or fake and the adversarial training loss $L_{D_z}^{adv}$ is defined as follows:

$$\min_{G,\mathcal{F},\mathcal{B},\mathcal{M}} \max_{D_z} \mathcal{L}_{D_z}^{adv} = \mathbb{E}_x[\log D_z(x)] + \mathbb{E}_z[\log(1 - D_z(\bar{x}_z))] + \mathbb{E}_{z,f,b}[\log(1 - D_z(\bar{x}))]. \tag{7}$$

**Controllable regularization for foreground-background subnetworks.** A mutual information regularization is incorporated into the $\mathcal{F}$ and $\mathcal{B}$ to associate the variables $f$ and $b$ with the synthesized $\bar{x}_f$ and $\bar{x}_b$. Similar to InfoGAN [29], we obtain similar results by maximizing the lower bound on the variance of the mutual information.

$$\max_{D_f, D_b} L_{info} = I(\bar{x}_f, f) + I(\bar{x}_b, b) \geq \mathbb{E}_{z,f,b}[\log D_f(f \mid \bar{x}_f) + \log D_b(b \mid \bar{x}_b)], \tag{8}$$

where $I(\cdot)$ is the mutual information, two discriminators $D_f$ and $D_b$ approximate posterior distribution of $p(f|\bar{x}_f)$ and $p(b|\bar{x}_b)$. As implemented in [37, 20, 25], the $I(\cdot)$ is regarded as a regularization of a class of unsupervised clusters. In other words, $D_f$ guides $\mathcal{F}$ to focus on generating foreground regions and learning the features of objects. The similar analysis applies to $D_b$. According to [10, 38], it is difficult for GANs to learn a set of disconnected manifolds. Intuitively, $\mathcal{F}$ and $\mathcal{B}$ learn simpler and continuous distributions, and thereby $L_{info}$ improves the quality of image generation.

## 3.3 The Segmentation Module of DS-ComGAN

The following two networks are defined: a segmentation network $\mathcal{S}$ predicts segmentation masks conditioned on the input images; a discriminator $D_m$ classifies masks as real or fake. Our key idea is to exploit the high semantic masks synthesized by disentanglement module to align the image distribution and the mask distribution via adversarial training.

**Image distribution alignment.** In the disentanglement module, $D_z$ is trained to identify $x$ sampled from $p_{data}$ and $\bar{x}$ composited by our model. When the model achieve Nash equilibrium, $\bar{x}$ has the identical distribution as $x$, i.e. $p_{data}(x) = p(\bar{x})$. Hence, image distribution alignment is accomplished.

**Mask distribution alignment.** In the first step, we input the real image $x$ and the synthetic image $\bar{x}$ into the segmentation network to obtain the corresponding prediction masks, that is, $x_m = \mathcal{S}(x)$ and $\hat{x}_m = \mathcal{S}(\bar{x})$. Then, a mask triplet $(\bar{x}_m, \hat{x}_m, x_m)$ is fed into the discriminator $D_m$ where the synthesized mask $x_m$ is regarded as real and both the predicted masks $\hat{x}$ and $x_m$ are regarded as fake. Mask distributions of $x_m$, $\hat{x}_m$ and $\bar{x}_m$ are aligned via adversarial training,

$$\min_{\mathcal{S}} \max_{D_m} \mathcal{L}_{D_m}^{adv} = \mathbb{E}_{\bar{x}_m, \hat{x}_m, x_m}[\log D_m(\bar{x}_m) + \log(1 - D_m(\hat{x}_m)) + \log(1 - D_m(x_m))]. \quad (9)$$

The mask distribution alignment makes the predicted masks clearer, but this adversarial training strategy causes the issue that predicted masks are inconsistent with the input images. Hence, the corresponding consistency loss $L_{cons}$ is introduced to ensure that the predicted masks are consistent with the inputs, rather than merely learning the mask distribution.

$$\min_{\mathcal{S}} \mathcal{L}_{cons} = \mathbb{E}_{\bar{x}, \bar{x}_m} [\|\bar{x}_m - \mathcal{S}(\bar{x})\|]. \quad (10)$$

We transform the unsupervised segmentation task into two adversarial tasks and acquire segmentation masks by enhancing image and mask distribution consistency.

## 3.4 Model Training

The whole training process of ComGAN is simple, with only an adversarial loss and a binary regularization. The learning objective is as follows:

$$L_{all} = \min_{\Phi, \mathcal{F}, \mathcal{B}, \mathcal{M}} \max_{D} \mathcal{L}_D^{adv} + \min_{\Phi, \mathcal{M}} \beta \mathcal{L}_{\text{binary}} , \quad (11)$$

where $\mathcal{L}_{\text{binary}}$ is the pixel-wise binary entropy of mask from [19]. An explicit formulation of $\mathcal{L}_{\text{binary}}$ is shown in Appendix B.2. The sharpness of the semantic mask is modified by fine-tuning $\beta$. The results of the visualization on fine-tuning $\beta$ are available in Appendix C.1. All the constituent networks of DS-ComGAN perform two kinds of unsupervised tasks: image disentanglement and object segmentation. Overall, we optimize the following loss:

$$L_{all} = \overbrace{\max_{D_f, D_b} L_{info}}^{\text{objective for image disentanglement task}} + \underbrace{\min_{G, \mathcal{F}, \mathcal{B}, \mathcal{M}} \max_{D_z} \mathcal{L}_{D_z}^{adv} + \min_{\mathcal{S}} \max_{D_m} \mathcal{L}_{D_m}^{adv} + \min_{\mathcal{S}} \lambda \mathcal{L}_{\text{cons}}}_{\text{objective for object segmentation task}} . \quad (12)$$

For the image disentanglement task, the model is performed in one stage. Then, the synthesized images and semantic masks are fed into the segmentation network. Hence, the objective segmentation process is composed of two stages.

## 4 Experiments

### 4.1 Experimental Setup

**Datasets and implementation details.** The experiments are conducted on five fine-grained image datasets and a multi-object dataset: CUB [39], FS-100 [40], Stanford-Cars [41]. Stanford-Dogs [41], Flowers [42], CLEVR6 [43]. The ground truth segmentation masks of Stanford-Cars and Stanford-Dogs are approximated by following the practice in [21, 44, 20]. More details about datasets and the implementation are available in the supplementary material.

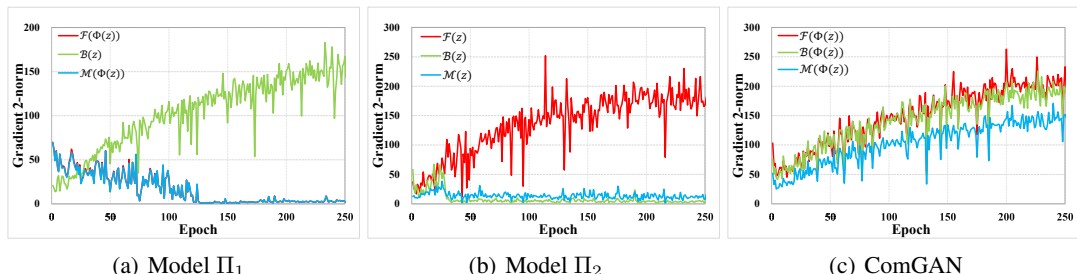

| (a) Model $\Pi_1$ | (b) Model $\Pi_2$ | (c) ComGAN |

Figure 4: **Comparison results of gradient 2-norm during the training.**

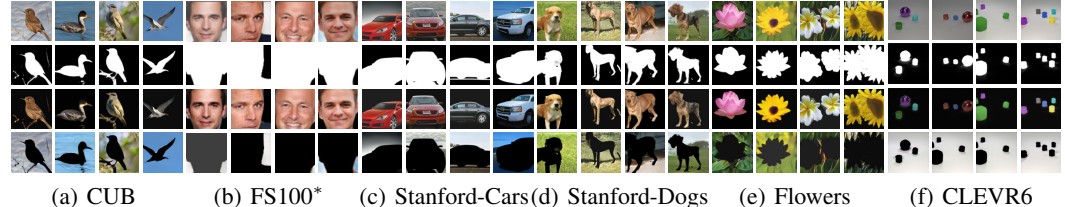

(a) CUB     (b) FS100*     (c) Stanford-Cars (d) Stanford-Dogs     (e) Flowers     (f) CLEVR6

Figure 5: **Qualitative generation results for each dataset**. Zoom in for better visibility. From top to bottom: (i) final images; (ii) masks; (iii) foregrounds; (iv) backgrounds. * indicates that we incorporate a binary regularization to obtain a high semantic mask, since the foreground almost covers the entire image.

**Baseline and evaluation protocol.** In order to analyze the effectiveness of the adopted components, we established a baseline model, i.e. SimpleGAN [25]. SimpleGAN aims to generate a realistic image, and its network contains a generator and a discriminator. For a fair comparison, we use SimpleGAN as the feature generator of ComGAN and DS-ComGAN. We assess synthesis quality in terms of Inception Score (**IS**) [45] and Fréchet Inception Distance (**FID**) [46], which are computed on 20K randomly synthesized images. To quantitatively evaluate the quality of the predicted masks, both the Intersecion of Union (**IoU**) and Dice score (**Dice**) [47] are used as the evaluation metrics.

### 4.2 Trivial Solutions Avoidance based on ComGAN

**Avoidance of trivial solutions.** To show that ComGAN solves trivial solutions, we select two typical networks represent model $\Pi_1$ and $\Pi_2$ for comparison. All models only include identical adversarial loss and binary regularization. As a typical network for model $\Pi_1$, FineGAN has two generator where a generates foreground and mask and another generates background. PerturbGAN is selected as the typical network for model $\Pi_2$, which consists of three generators that generates foreground, background and mask, respectively. For a fair comparison, we make the following two things: (i) we use the code provided by the authors in FineGAN and retrain it with the identical training parameters; (ii) we simplify StyleGAN [48] with SimpleGAN in PerturbGAN and retrain it. As shown in Fig. 2, the mask synthesized by ComGAN avoids convergence to trivial solutions and has non-trivial semantic information. In addition, we track the gradient 2-norm of the three models. It is clear from Fig. 4 that the gradient norms of mask networks in the model $\Pi_1$ and $\Pi_2$ converge to zero and both models degrade to an original GAN, while our method effectively avoids vanishing gradients.

### 4.3 Image Disentanglement based on DS-ComGAN

**Quantitative evaluation of image generation.** We compare DS-ComGAN with the competing GAN-based generative models without any advanced GAN training strategy. As illustrated in Table 1, the results show that DS-ComGAN significantly outperforms all the SOTA weakly supervised and semi-supervised methods. FineGAN and MixNMatch may be limited by the foreground-background conflict, such as trivial masks. SSC-GAN might have trouble in learning a set of disconnected manifolds. To emphasize that DS-ComGAN is fully unsupervised and adapts to complex multi-object scenarios, we further perform our model on diverse datasets. Stanford-Dogs and Flowers typically include a single foreground object, whereas CLEVR6 contains a variety of occluded and truncated objects. Table 3 shows that our model is robust on various datasets and outperforms the SOTA image

**Varying variable z**

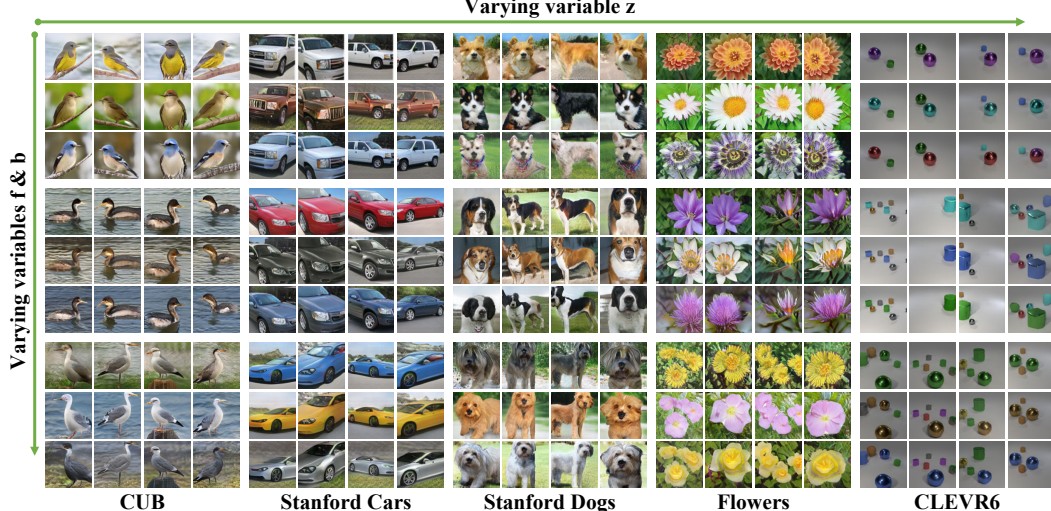

Varying variables f & b

| CUB | Stanford Cars | Stanford Dogs | Flowers | CLEVR6 |

Figure 6: **The controllable image generation via varying variables $f$, $b$ and $z$.** Zoom in for better visibility.

| Methods | Sup. | CUB | | FS-100 | | Stanford-Cars | |
|---|---|---|---|---|---|---|---|
| | | FID ↓ | IS↑ | FID ↓ | IS↑ | FID ↓ | IS↑ |
| Triple-GAN [49] | Semi. | 140.94 | 3.94±0.06 | 91.05 | 1.45±0.03 | 114.12 | 2.45±0.06 |
| EnhancedTGAN [40] | Semi. | 133.57 | 4.17±0.06 | 57.58 | 1.57±0.02 | 105.20 | 2.43±0.05 |
| Triangle-GAN [50] | Semi. | 96.42 | 4.36±0.05 | 35.49 | 1.71±0.04 | 61.44 | 2.77±0.10 |
| $R^3$-CGAN [51] | Semi. | 88.62 | 4.43±0.06 | 25.28 | 1.73±0.02 | 44.57 | 3.05±0.04 |
| SSC-GAN§ [27] | Semi. | 20.03 | 4.68±0.04 | 20.65 | 1.82±0.03 | 39.02 | **3.10±0.03** |
| FineGAN§ [25] | Weak. | 46.68 | 4.62±0.03 | 24.63 | 1.76±0.02 | 45.72 | 2.85±0.04 |
| MixNMatch§ [31] | Weak. | 45.59 | 4.78±0.08 | 25.63 | 1.71±0.05 | 45.94 | 2.60±0.05 |
| SN-GAN [52] | Unsup. | 160.09 | 4.21±0.05 | 41.26 | 1.66±0.05 | 53.20 | 2.80±0.05 |
| DS-ComGAN§ | Unsup. | **16.26** | **4.79±0.47** | **20.15** | **1.83±0.32** | **34.17** | 2.84±0.12 |

Table 1: **Image synthesis results for each dataset measured in FID and IS.** DS-ComGAN is compared with the state-of-the-art un(semi-)supervised GAN-based models. § indicates that the models have the ability to achieve image disentanglement.

disentanglement methods. Fig 5 shows the qualitative generation results for each dataset, from which it is evident that our model generates not only realistic images but also clear semantic masks.

**Controllable image disentanglement.** Fig. 6 illustrates that by varying three independent variables, DS-ComGAN synthesizes realistic and diverse images. This indicates that the model achieves controlled image disentanglement. We see that variable $z$ is associated with the masks. The pose of the object is altered with the varying variable $z$. The colors and textures of the foreground and background are associated with variables $f$ and $b$. To clarify the role of $f$ and $b$ in image disentanglement, Fig 7 further analyzes the disentanglement of foreground and background. In summary, DS-ComGAN associates three independent variables ($f$, $b$ and $z$) with the foreground, background and mask, respectively.

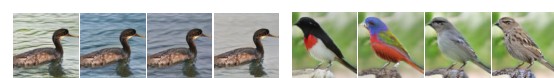

(a) Varying $b$, fixed $f$ and $z$  (b) Varying $f$, fixed $b$ and $z$

Figure 7: **Disentanglement of foreground-background.**

| Method | FID ↓ | IS ↑ |
|---|---|---|
| Baseline | 89.87 | 4.45±0.46 |
| + Independent subnetworks. | 21.35 | 4.75±0.08 |
| + $L_{info}$-based Reg. | 17.32 | 4.73±0.08 |
| + Extract global features. | 16.26 | 4.79±0.47 |
| *Improvement* | **73.61** | **0.34** |

Table 2: **The results of the baseline and variants on CUB.**

| Methods | Single Object | | | | Multi-Object | |
|---|---|---|---|---|---|---|
| | Stanfor-Dogs | | Flowers | | CLEVR6 | |
| | FID ↓ | IS↑ | FID ↓ | IS↑ | FID ↓ | IS↑ |
| SSC-GAN[§] [27] | 64.26 | 8.97±0.12 | 29.09 | 3.41±0.03 | \ | \ |
| FineGAN[§] [25] | 69.52 | 8.27±0.17 | \ | \ | \ | \ |
| MixNMatch[§] [31] | 68.31 | 8.32±0.06 | \ | \ | \ | \ |
| DS-ComGAN[§] | **60.84** | **9.17±0.23** | **27.19** | **3.42±0.04** | 77.08 | 2.75±0.05 |

Table 3: **Performance of DS-ComGAN on datasets with various attributes.** Noting that the Flowers dataset lacks bounding box annotation, FineGAN and MixNMatch both are unsuitable for this dataset (marked as \ ). CLEVR6 lacks bounding box annotation and labels. Consequently, only our model is suitable for CLEVR6.

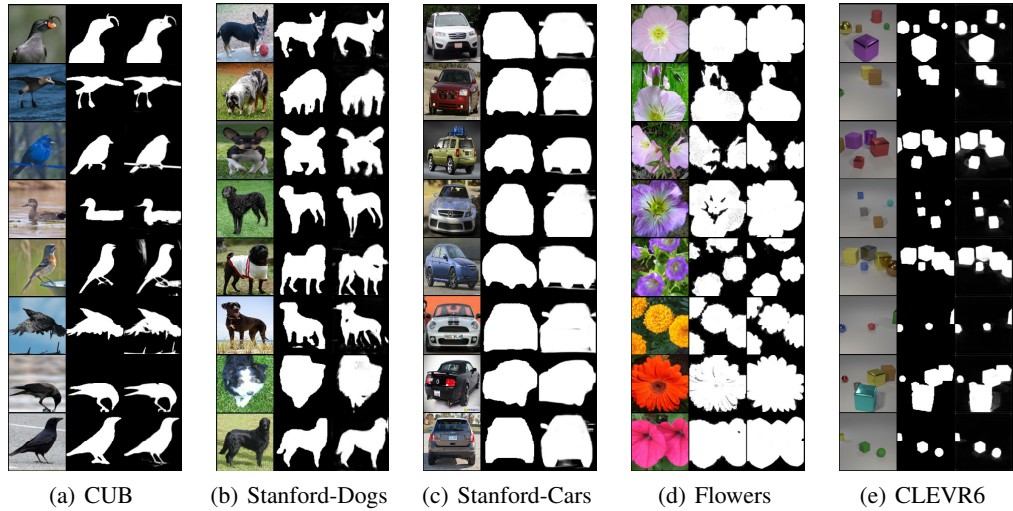

|  (a) CUB  |  (b) Stanford-Dogs  |  (c) Stanford-Cars  |  (d) Flowers  |  (e) CLEVR6  |

Figure 8: **Qualitative segmentation results for each dataset.** From left to right: (i) observed images, (ii) ground-truth masks, (iii) predicted masks. More samples can be found in the supplementary material.

**Effectiveness of model components** To evaluate the contribution of each proposed component, we analyze quantitatively the performance margin between the baseline model and ComGAN. As demonstrated in Table 2, the incorporation of $\mathcal{F}$, $\mathcal{B}$ and $\mathcal{M}$ significantly improves the performance of the baseline model. It illustrates that ComGAN has an excellent image generation performance. As analyzed in section 3.2, the $L_{info}$ causes a decrease in FID by about 4 points.

## 4.4    Unsupervised Segmentation based on DS-ComGAN

To demonstrate that our method applies to complex real-world domains and multi-object scenarios with challenging spatial configurations, along the lines of [44], DS-ComGAN performs the unsupervised segmentation task on the various datasets. Then, we perform a comparison between DS-ComGAN and several state-of-the-art un(weakly-)supervised segmentation methods.

**Qualitative analysis.** In Fig. 8, qualitative segmentation results of DS-ComGAN are presented. We observe that the predicted masks segment visual details more precisely than the ground-truth masks, such as the legs of birds and the rearview mirrors of cars. Although the predicted masks properly detect and segment foreground objects on the Flowers and Stanford-Cars datasets, the masks are inconsistent with the foreground images, which is mitigated by increasing $\lambda$.

**Quantitative comparisons to prior work.** Table 4 demonstrates that DS-ComGAN exhibit robustness to diverse datasets and achieve superior performance across all metrics and datasets. This implies that the model pays more attention to the discrepancy between the objects and background rather than the number of objects. In addition, the result benefits from the images and high semantic masks synthesized by DS-ComGAN, which ensure two high-quality distribution alignments.

| | | | | | | | | | | |
|---|---|---|---|---|---|---|---|---|---|---|
| | **Single Object** | | | | | | | | **Multi-Object** | |
| | CUB | | Stanford-Dogs | | Stanford-Cars | | Flowers | | CLEVR6 | |
| Methods | IoU↑ | Dice↑ | IoU↑ | Dice↑ | IoU↑ | Dice↑ | IoU↑ | Dice↑ | IoU↑ | Dice↑ |
| W-Net [34] | 24.8 | 38.9 | 47.7 | 62.1 | 52.8 | 67.6 | - | - | - | - |
| GrabCut [53] | 30.2 | 42.7 | 58.3 | 70.9 | 61.3 | 73.1 | 69.2 | 79.1 | 19.0 | 30.5 |
| ReDO†✳ [35] | 46.5 | 60.2 | 55.7 | 70.3 | 52.5 | 68.6 | 76.4 | - | 18.6 | 31.0 |
| OneGAN◇✳ [21] | 55.5 | 69.2 | 71.0 | 81.7 | 71.2 | 82.6 | | - | - | - |
| IODINE† [54] | 30.9 | 44.6 | 54.4 | 67.0 | 51.7 | 67.3 | - | - | 19.9 | 32.4 |
| PerturbGAN [11] | 38.0 | - | - | - | - | - | - | - | - | - |
| Slot-Attn.† [55] | 35.6 | 51.5 | 38.6 | 55.3 | 41.3 | 58.3 | - | - | 83.6 | 90.7 |
| IEM+SegNet [36] | 55.1 | 68.7 | - | - | - | - | 76.8 | **84.6** | - | - |
| DRC [44] | 56.4 | 70.9 | 71.7 | 83.2 | 72.4 | 83.7 | - | - | 84.7 | 91.5 |
| DS-ComGAN | **60.7** | **71.3** | **74.5** | **84.6** | **76.7** | **86.6** | **76.9** | 83.1 | **90.0** | **94.6** |

Table 4: **Segmentation results on training data measured in IoU and Dice**. DS-ComGAN is compared with the state-of-the-art un(weakly-)supervised segmentation methods. Following the [44], † indicates unfair baseline results obtained using extra ground-truth information. ✳ represents a GAN-based model. OneGAN◇ is a weakly supervised baseline, which requires clean backgrounds as additional inputs.

## 5   Conclusion

As the limitation of method, DS-ComGAN has struggled to achieve the desired performance when the foreground object features are highly diverse (e.g., HKU-IS [56]). We conjecture that the limitation might relate to the capacity of model. DS-ComGAN performs excellently in controlled image synthesis tasks, which may cause the incidence of image falsification.

We first analyze that the source of trivial solutions is vanishing gradients on the mask. Subsequently, we propose ComGAN, a simple unsupervised generative model, which simultaneously generates realistic images and high semantic masks, and effectively avoids trivial solutions. Then, we design DS-ComGAN based on ComGAN, which exhibits excellent performance in both image disentanglement and unsupervised segmentation tasks. Notably, DS-ComGAN obtains these excellent performances by relying on its architecture rather than complex and clever regularization. It implies that ComGAN has the potential to shine in many tasks.

## Acknowledgments

This work was supported by the Natural Science Foundation of China under Grant 62076255 and 62177047; Open Research Projects of Zhejiang Lab (NO. 2022RC0AB07); Hunan Provincial Science and Technology Plan Project 2020SK2059; Key projects of Hunan Education Department 20A88; National Science Foundation of Hunan Province 2021JJ30082, 2019JJ20025 and 2019JJ40406; National Social Science Fund of China (No. 20&ZD120).

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
