# A  Details on Datasets and Hyperparameters

## A.1  Dataset Details

**Caltech-UCSD Birds-200-2011.** This dataset contains about 12K bird images of 200 classes annotated with bounding boxes. The ground truth segmentation masks of the foreground object are annotated by humans and provided in the official release [39]. As in the previous work [25, 31, 44], We use the provided bounding box to extract a center square from the image, and scale it to $128 \times 128$ pixels.

**Stanford Dogs.** This dataset consists of about 20K dog images of 120 classes annotated with bounding boxes [41]. Similar to [25, 31, 44], We use the provided bounding box to extract a center square from the image, and scale it to $128 \times 128$ pixels. Since the ground-truth segmentation masks are not provided, we adopt the approximate ground-truth masks, generated by Yu et al. [44]. The specific generation process is as follows. Yu et al. follow the practice in Benny and Wolf[21] to approximate ground-truth masks with Mask R-CNN[57], pre-trained on the COCO[58] dataset with a ResNet-101[59] backend. The pre-trained model is acquired from the detectron2 [60] toolkit. Along the filtering strategy of [44], there exist 5,024 images with a clear foreground-background setup and high-quality mask.

**Stanford Cars.** This dataset consists of about 16K car images of 196 classes annotated with bounding boxes [41]. We use the provided bounding box to extract a center square from the image, and scale it to $128 \times 128$ pixels. To approximate the ground-truth masks, we adopt a similar process and filtering strategy to the Stanford Dogs dataset. Finally, there exist 12,322 images for experimentation.

**FS-100.** This dataset consists of about 15K human face images from the 100 largest classes [40, 27] of FaceScrub[45]. FaceScrub comprises a total of 107,818 face images of 530 celebrities. For a fair comparison, we follow the [27] by scaling all the images to $64 \times 64$ pixels.

**Flowers.** This dataset consists of about 8K flower images of 102 classes [42]. The ground truth segmentation masks are created by an automated approach designed exclusively for segmenting flowers in color photographs [61]. We scale all the images to $128 \times 128$ pixels.

**CLEVR6** This dataset is a subset of the CLEVR dataset [43] with masks generated by Greff et al. [54]. We take the first 8K samples from CLEVR and scale these samples to $128 \times 128$ pixels.

## A.2  Hyperparameters and Training Details.

We adopt the ADAM optimizer[62] with a learning rate of $\varsigma = 2 \times 10^{-4}$ and momentum parameters of $\beta_1 = 0.5$ and $\beta_2 = 0.999$. Following FineGAN[25] and DRC[44], we augment the real image with a horizontal flip and a random resizing crop. The variable dimensions are set as follows: (1) CUB: $N = 200$; (2) FS-100: $N = 100$; (3) Stanford-Dogs: $N = 150$; (4) Stanford-Cars: $N = 200$; (5) Flowers: $N = 150$; (6) CLEVR6: $N = 20$. We empirically set $\lambda = 5$ defined in equation (12) for all datasets. The maximum training epochs are set to 500, where the disentanglement module is first trained for 450 epochs with a batch size of 32, and the segmentation module is second trained for 50 epochs with a batch size of 64. All experiments are performed under the environments of NVIDIA GeForce RTX 3090 GPUs and an Intel Xeon Gold 6130 CPU.

# B  Details on Proofs and Regularization

## B.1  Proofs of things

*Proof of Lemma 1:* (Sufficiency) Under Assumption 1, the composes image $\bar{x}$ at each pixel $i, j \in \mathbb{R}$ can be written as follows:

$$\bar{x}[i,j] = \bar{x}_f[i,j] \cdot \bar{x}_m[i,j] + \bar{x}_b[i,j] \cdot (1 - \bar{x}_m[i,j]), \tag{13}$$

where $\bar{x}_f$ is the foreground and $\bar{x}_b$ is the background. As the model converges to the first trivial solution, the composite image degenerates to the foreground or background, i.e. $\bar{x}[i,j] = \bar{x}_f[i,j]$ or $\bar{x}[i,j] = \bar{x}_b[i,j]$. It implies that the compositional generation process degenerates into a single foreground or background generation, which causes vanishing gradients on the mask.

Let $\bar{G}_{\theta_m}$ be masks generator and $\theta_m$ be the network parameters of $\bar{G}_{\theta_m}$. Then, by using the chain rule, it can be obtained that

$$\nabla_{\theta_m} L_{all} = \frac{\partial L_{all}}{\partial \bar{x}[i,j]} \cdot \frac{\partial \bar{x}[i,j]}{\partial \bar{x}_m[i,j]} \cdot \nabla_{\theta_m} \bar{G}_{\theta_m}[i,j]$$
$$= \frac{\partial L_{all}}{\partial \bar{x}[i,j]} \cdot (\bar{x}_f[i,j] - \bar{x}_b[i,j]) \cdot \nabla_{\theta_m} \bar{G}_{\theta_m}[i,j]. \tag{14}$$

As the model converges to the second trivial solution, the foreground and background are identical, i.e. $\bar{x}_f[i,j] = \bar{x}_b[i,j]$, which causes vanishing gradients on the mask.

(Necessity) Vanishing gradients on the mask means that no gradient is passed to the mask generator. The synthesized mask is meaningless and fails to achieve foreground-background disentanglement. Therefore, the model converges to the first trivial solution when the synthesized mask is full or empty. In other cases, the model converges to the second trivial solution. $\square$

*Proof of Theorem 1:* $\|\nabla_\theta L_D^{adv}\|_2$ can be written as

$$\left\| \nabla_\theta \mathbb{E}_{z \sim p(z)} \left[ \log \left( 1 - D \left( G_\theta(z) \right) \right) \right] \right\|_2 \geq \sigma. \tag{15}$$

For simplicity of the following analysis, we assume the model converges to the first trivial solution and $\bar{x}_m = 1$. Under Assumption 1, it can be obtained that $\bar{x} = \bar{x}_f$ and $\partial L_{all}/\partial \bar{x}_m = \partial L_{all}/\partial \bar{x}_b = 0$. Hence, $\nabla_\theta L_{all}$ can be decomposed as $\nabla_{(\theta_\phi, \theta_f)} L_{all}$, and the gadient norms (15) can be written as:

$$\left\| \nabla_{\theta_\phi} \mathbb{E}_{z \sim p(z)} \left[ \log \left( 1 - D \left( \mathcal{F}(\Phi(z)) \right) \right) \right] \right\|_2^2 + \left\| \nabla_{\theta_f} \mathbb{E}_{\phi \sim p(\Phi(z))} \left[ \log \left( 1 - D \left( \mathcal{F}_{\theta_f}(\phi) \right) \right) \right] \right\|_2^2 \geq \sigma^2. \tag{16}$$

Along the lines from [63, 64], we use Jensen's inequality and the chain rule:

$$\left\| \nabla_{\theta_f} \mathbb{E}_{\phi \sim p(\Phi(z))} \left[ \log \left( 1 - D \left( \mathcal{F}_{\theta_f}(\phi) \right) \right) \right] \right\|_2^2 \leq \mathbb{E}_{\phi \sim p(\Phi(z))} \left[ \frac{\left\| \nabla_{\theta_f} D \left( \mathcal{F}(\phi) \right) \right\|_2^2}{\left| 1 - D \left( \mathcal{F}(\phi) \right) \right|^2} \right]$$

$$\leq \mathbb{E}_{\phi \sim p(\Phi(z))} \left[ \frac{\left\| \nabla_{\bar{x}} D \left( \mathcal{F}(\phi) \right) \right\|_2^2 \left\| J_{\theta_f} \mathcal{F}(\phi) \right\|_2^2}{\left| 1 - D \left( \mathcal{F}(\phi) \right) \right|^2} \right]$$

$$< \mathbb{E}_{z \sim p(z)} \left[ \frac{\left( \left\| \nabla_x D^* \left( G^*(z) \right) \right\|_2 + \epsilon \right)^2 \left\| J_{\theta_f} \mathcal{F}(\Phi(z)) \right\|_2^2}{\left( \left| 1 - D^* \left( G^*(z) \right) \right| - \epsilon \right)^2} \right]$$

$$\leq \frac{\delta^2 \epsilon^2}{(1/2 - \epsilon)^2}. \tag{17}$$

Then, it follows that

$$\left\| \nabla_{\theta_\phi} \mathbb{E}_{z \sim p(z)} \left[ \log \left( 1 - D \left( \mathcal{F}(\Phi(z)) \right) \right) \right] \right\|_2^2 > \sigma^2 - \frac{\delta^2 \epsilon^2}{(1/2 - \epsilon)^2}. \tag{18}$$

Notice that the mask can be written as $\mathcal{M}(\Phi(z))$, which means

$$\left\| \nabla_{(\theta_\phi, \theta_m)} \mathbb{E}_{z \sim p(z)} \left[ \log \left( 1 - D \left( \mathcal{F}(\Phi(z)) \right) \right) \right] \right\|_2^2 > \sigma^2 - \frac{\delta^2 \epsilon^2}{(1/2 - \epsilon)^2}. \tag{19}$$

A similar discussion can be applied to the case of $\bar{x}_m = 0$ $\square$

*Proof of Corollary 1:* By Theorem 1, the larger the value of $\rho = \sigma^2 - \frac{\delta^2 \epsilon^2}{(1/2 - \epsilon)^2}$, the easier it is for the model to escape from trivial solutions. We observe that the $\delta^2$ is decreased by reducing the parameter of $\theta_f$ and $\theta_b$ from the inequality $\max\{\mathbb{E}_{\phi \sim p(\Phi(z))} \|J_{\theta_f} \mathcal{F}(\phi)\|_2^2, \mathbb{E}_{\phi \sim p(\Phi(z))} \|J_{\theta_b} \mathcal{B}(\phi)\|_2^2\} \leq \delta^2$. It is notable that if the $\theta_f$ and $\theta_b$ have few parameters or even no parameters, our model degrades into an original GAN. Therefore, the $\mathcal{F}$ and $\mathcal{B}$ should be designed as lightweight modules (e.g. adding residual connections), so as to enhance the capacity of the module with fewer parameters. Furthermore, the $\rho$ is the gradient norm from the decoder $\Phi$. It is a natural idea to increase $\rho$ by raising the capacity of $\Phi$. In addition, the update of $\Phi$ does not necessarily mean the update of $\mathcal{M}$. If $\mathcal{M}$ is a deep neural network, it may not be able to map the fluctuations of features to the mask space, that is, $\mathcal{M}(\phi) \approx \mathcal{M}(\phi + \triangle\phi)$. Therefore, $\mathcal{M}$ should be designed as a shallow neural network. For example, when $\mathcal{M}$ is a sigmoid layer, it is easy to get that $\mathcal{M}(\phi) \neq \mathcal{M}(\phi + \triangle\phi)$ by the monotonicity of the sigmoid function. As for the second trivial solution, $\bar{x}_f = \bar{x}_b$ is a fragile equilibrium. We can break the identical mapping $\mathcal{F}_{\theta_f}(\phi) = \mathcal{B}_{\theta_b}(\phi)$, by changing parameters of $(\theta_f, \theta_b)$ or modifying the structure of the $\mathcal{F}$ and $\mathcal{B}$. $\square$

## B.2  Regularization

The binary regularization $\mathcal{L}_{\text{binary}}$ enforces the output to be close to either 0 or 1, which is as follows:

$$\mathcal{L}_{\text{binary}} = \sum_{i=1}^{N} -m_i \log_2(m_i) - (1-m_i) * \log_2(1-m_i), \tag{20}$$

## C  Additional results about ComGAN

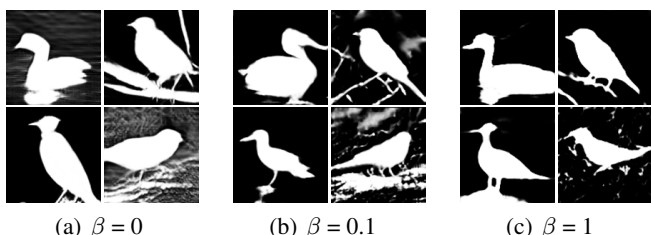

(a) $\beta = 0$      (b) $\beta = 0.1$      (c) $\beta = 1$

Figure 9: **Visual comparisons of mask generation by fine-tuning $\beta$.**

### C.1  Qualitative evaluation of mask synthesis.

We compare the visual effects of masks synthesized by ComGAN under various $\beta$. The comparison results are illustrated in Fig. 9. Without any supervised information, it is difficult for the model to distinguish which part of the image belongs to the foreground or the background. Therefore, we introduce the binary regularization $\beta L_{binary}$ to separate the foreground-background regions.

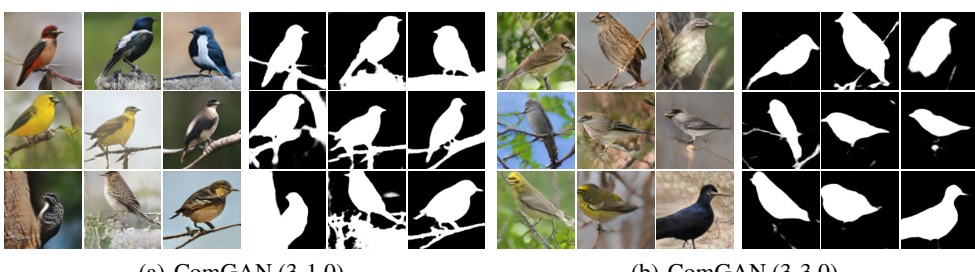

(a) ComGAN (3-1,0)            (b) ComGAN (3-3,0)

Figure 10: **Visual comparisons of ComGAN variants with imbalanced foreground and background subnetworks.** For each subfigure: (left) synthesized images, (right) corresponding masks.

### C.2  Imbalance of foreground and background subnetworks

In contrast to the background, the foreground is not occluded and contains complete information. Therefore, we modify the foreground and background subnetworks with identical structures to make the two subnetworks have unbalanced network capacity. For simplicity, we define a ComGAN variant called ComGAN ($N_F - N_B, N_M$), where $N_F$, $N_B$ and $N_M$ denote the number of ResBlocks in the foreground, background and mask subnetworks, respectively. As shown in Fig. 10, a sizeable visual gap exists between the masks produced by ComGAN(3-1,0) and ComGAN (3-3,0). The masks in Fig. 10(a) contain more foreground information, such as branches and rocks, even if the information is insignificant. In contrast, the masks Fig. 10(b) lack partial details of objects, especially the bird's feet. It seems that we obtain higher-quality segmentation masks by trade-offs between the capacity of the foreground and background subnetworks.

### C.3  Relaxing constraints on mask subnetwork

Corollary 1 requires that the $\mathcal{M}$ is a shallow network. The property is a strong constraint on the network design. Therefore, we show that a ComGAN variant avoids trivial solutions even when

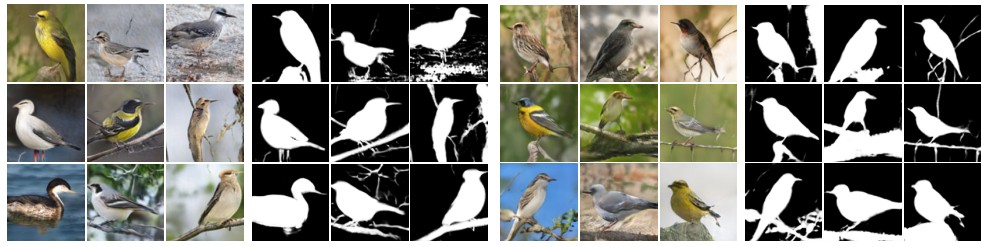

| (a) ComGAN (3-3,1) | (b) ComGAN (3-3,2) |

Figure 11: **Visual comparisons of ComGAN variants with various mask subnetworks.** For each subfigure: (left) synthesized images, (right) corresponding masks.

| Methods | Sup. | FID ↓ | IS↑ |
|---|---|---|---|
| Triple-GAN [49] | Semi. | 140.94 | 3.94±0.06 |
| EnhancedTGAN [40] | Semi. | 133.57 | 4.17±0.06 |
| Triangle-GAN [50] | Semi. | 96.42 | 4.36±0.05 |
| $R^3$-CGAN[51] | Semi. | 88.62 | 4.43±0.06 |
| SSC-GAN$^\S$ [27] | Semi. | **20.03** | 4.68±0.04 |
| FineGAN$^\S$[25] | Weak. | 46.68 | 4.62±0.03 |
| MixNMatch$^\S$ [31] | Weak. | 45.59 | **4.78±0.08** |
| SN-GAN [52] | Unsup. | 160.09 | 4.21±0.05 |
| ComGAN(3-3,0) | Unsup. | 21.72 | 4.73±0.08 |
| ComGAN(3-3,1) | Unsup. | 20.41 | 4.76±0.11 |
| ComGAN(3-3,2) | Unsup. | 20.80 | 4.68±0.07 |

Table 5: **Image synthesis results on CUB.** ComGAN variants are compared with the state-of-the-art GAN-based models in image synthesis.

| Methods | IoU↑ | Dice↑ |
|---|---|---|
| W-Net [34] | 24.8 | 38.9 |
| GrabCut [53] | 30.2 | 42.7 |
| ReDO†* [35] | 46.5 | 60.2 |
| OneGAN◇* [21] | 55.5 | 69.2 |
| IODINE† [54] | 30.9 | 44.6 |
| Slot-Attn.† [55] | 35.6 | 51.5 |
| IEM+SegNet [36] | 55.1 | 68.7 |
| DRC [44] | 56.4 | **70.9** |
| ComGAN (3-3,0) | 50.1 | 64.6 |
| ComGAN (3-3,1) | **56.9** | 69.8 |
| ComGAN (3-3,2) | 53.9 | 67.5 |

Table 6: **Segmentation results on CUB.** ComGAN variants are compared with the state-of-the-art un(weakly-)supervised segmentation methods.

the mask subnetwork is not a sigmoid function. ComGAN variants are designed by increasing the number of ResBlocks in the mask subnetwork. It can be seen from Fig. 10(b) and Fig 11 that the extension of the mask subnetwork does not result in trivial solutions and provides masks that retain more details.

## C.4   Quantitative evaluation of image generation.

As summarized in Table 5, ComGAN variants can achieve comparable FID scores. We conjecture that improved image synthesis quality is attributable to ComGAN's network architecture. The mask subnetwork and binary regularization force foreground and background subnetworks to focus on learning simpler and continuous foreground and background regions. Furthermore, the experimental results on GAN-based image composition models such as FineGAN and MixNMatch indicate that trivial solutions degrade the quality of image generation. Specifically, the first trivial solution degrades the generation process of composing the foreground and background to a single generation. The second trivial solution induces interference between the foreground and background, decreasing the composite image quality.

## C.5   Quantitative evaluation of segmentation masks.

To highlight that ComGAN variants generate high semantic masks, we obtain the predicted segmentation masks by the method in Section 3.3. As summarized in Table 6, ComGAN variants are simple and exhibit comparable performance with only two regularizations. Possibly limited by the known drawbacks of GANs such as unstable training, ComGAN variants have differences in segmentation results. Moreover, we observe that the quality of the predicted segmentation masks is improved when the image distribution is closer (i.e. the lower FID scores).

| | $N$=100 | $N$=150 | $N$= 200 | $N$=250 | $N$=300 |
|---|---|---|---|---|---|
| FID ↓ | 16.94 | 17.33 | **16.26** | 16.71 | 17.39 |
| IS ↑ | 4.63 | 4.71 | **4.79** | 4.76 | 4.69 |

Table 7: **Robustness results for variable dimensions.**

# D  Additional discussion about DS-ComGAN

## D.1  Influence of dimension of variables

Table 7 shows the FID and IS of DS-ComGAN trained on the CUD by varying the dimension of variables $N$. We observe that our model achieves comparable performance with various $N$. As discussed in section 3.2, the $N$ can be considered as the number of categories in clusters. Therefore, when $N$ equals 200, model performance is marginally improved.

## D.2  Ablation study on global feature extraction

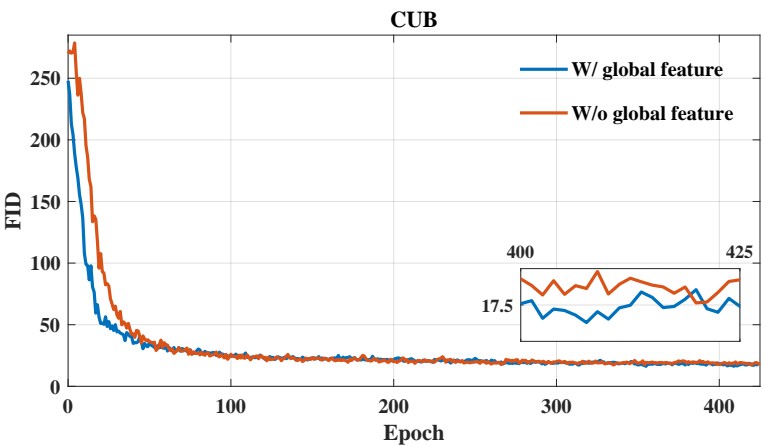

Figure 12: **The FID curves of training DS-ComGAN w/ or w/o global feature extraction.**

As shown in Fig. 12, the model performs global feature extraction on the CUB dataset, which leads to a rapid convergence and decrease of 1 to 2 points in the FID scores. Of note is that the extraction of global features consumes a large amount of GPU video memory and slightly improves the quality of image generation. However, the operation is necessary for specific datasets such as Stanford Cars, because it permits the mask to capture the entire distinction between the foreground and background, as illustrated in Fig. 13.

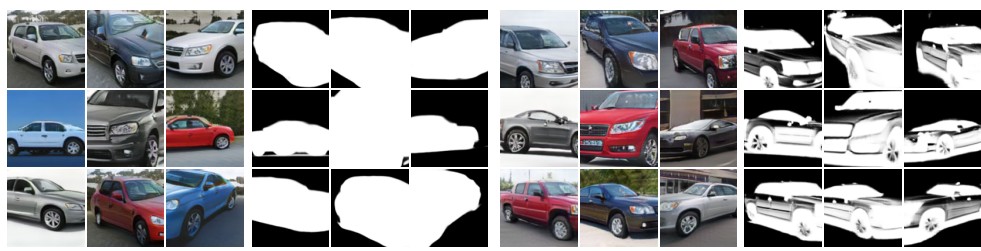

(a) Image synthesis w/ global feature            (b) Image synthesis w/o global feature

Figure 13: **Ablation study on extraction of global features.** For each subfigure: (left) synthesized images, (right) corresponding masks.

## D.3 Additional unsupervised segmentation results

See Fig. 14, Fig. 15, Fig. 16, Fig. 17, and Fig. 18 for more unsupervised segmentation results by our model on each dataset. From top to bottom, we display: (i) observed images, (ii) ground-truth masks, and (iii) predicted masks in each figure.

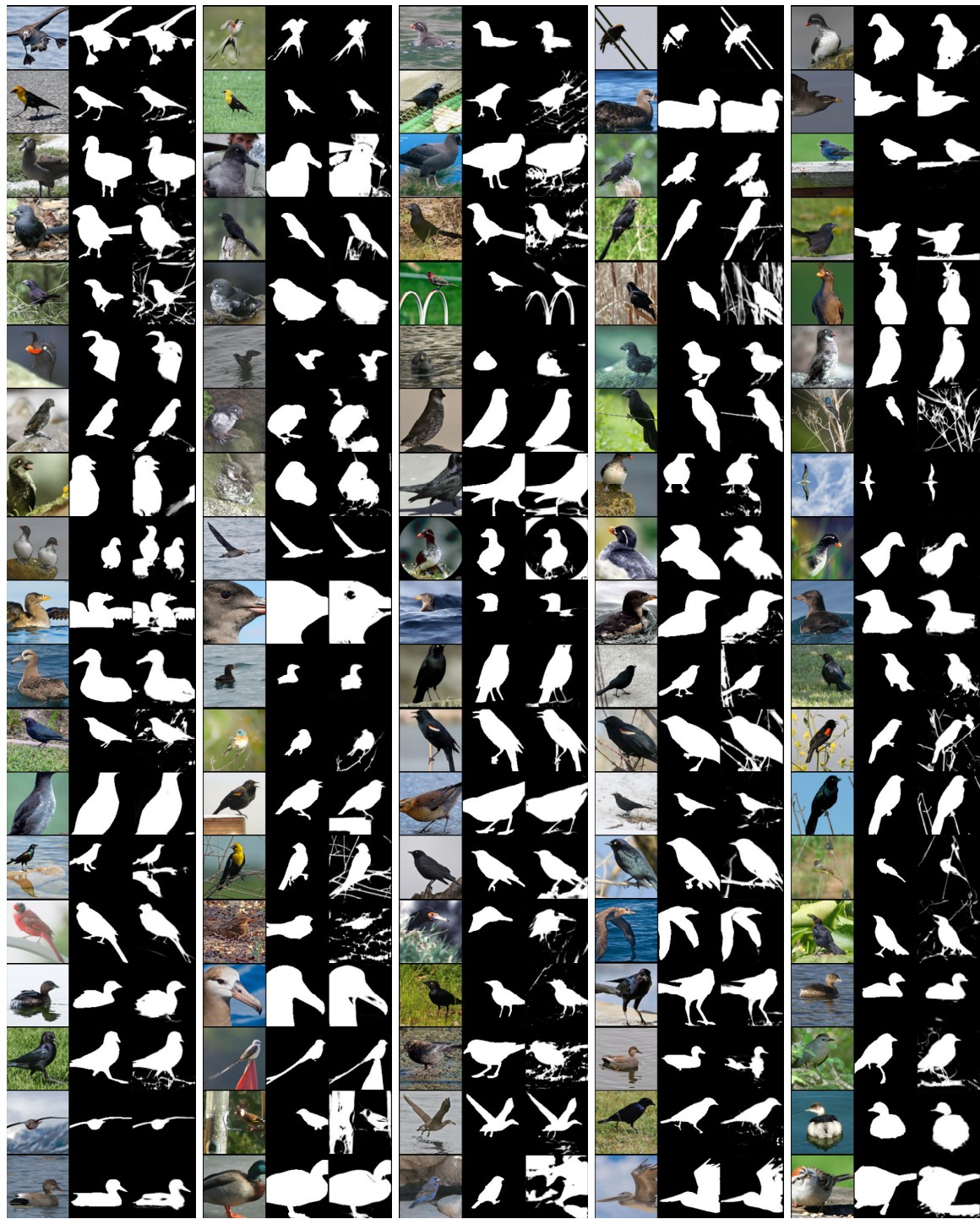

Figure 14: More randomly sampled segmentation masks on CUB.

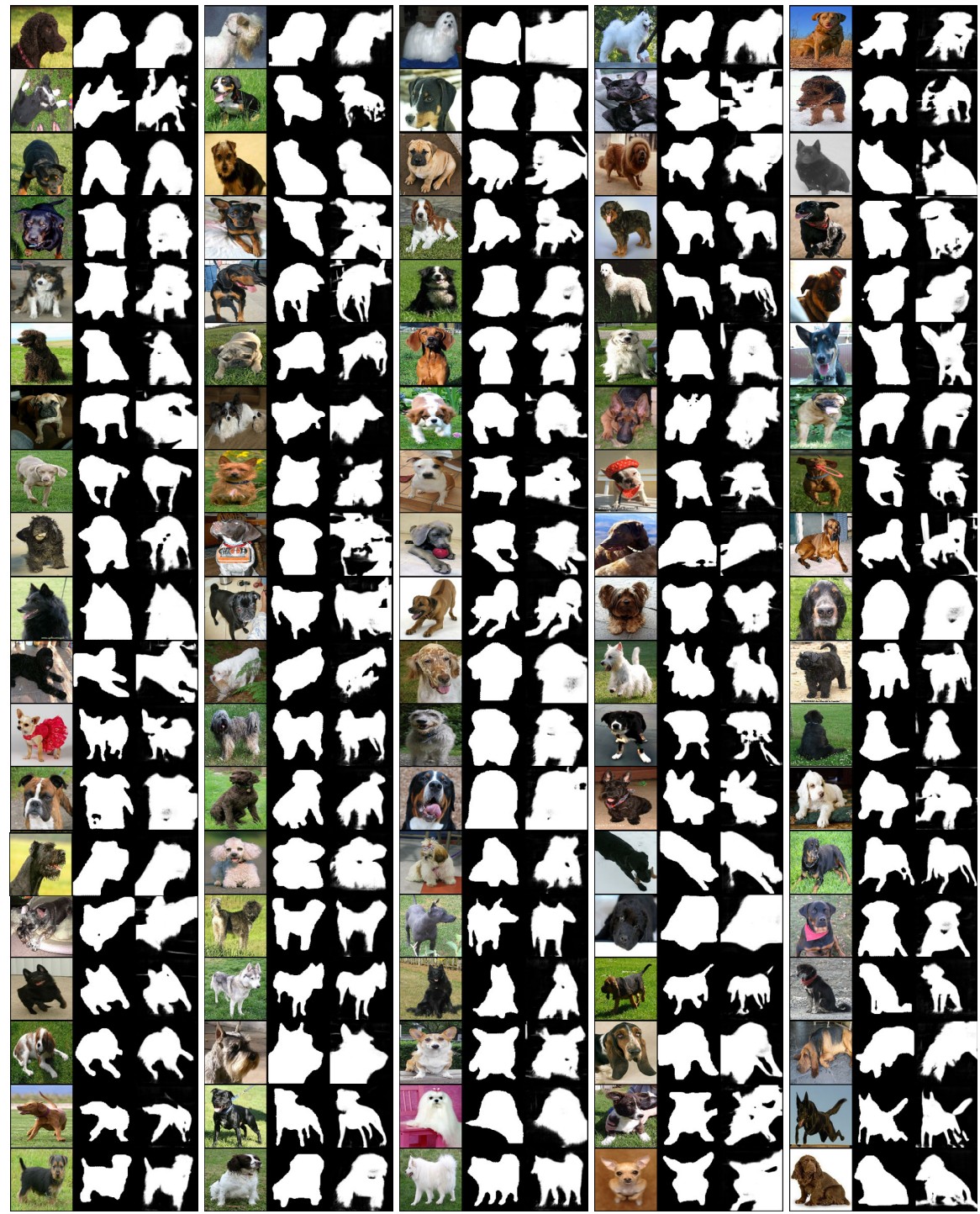

Figure 15: More randomly sampled segmentation masks on Stanford Dogs.

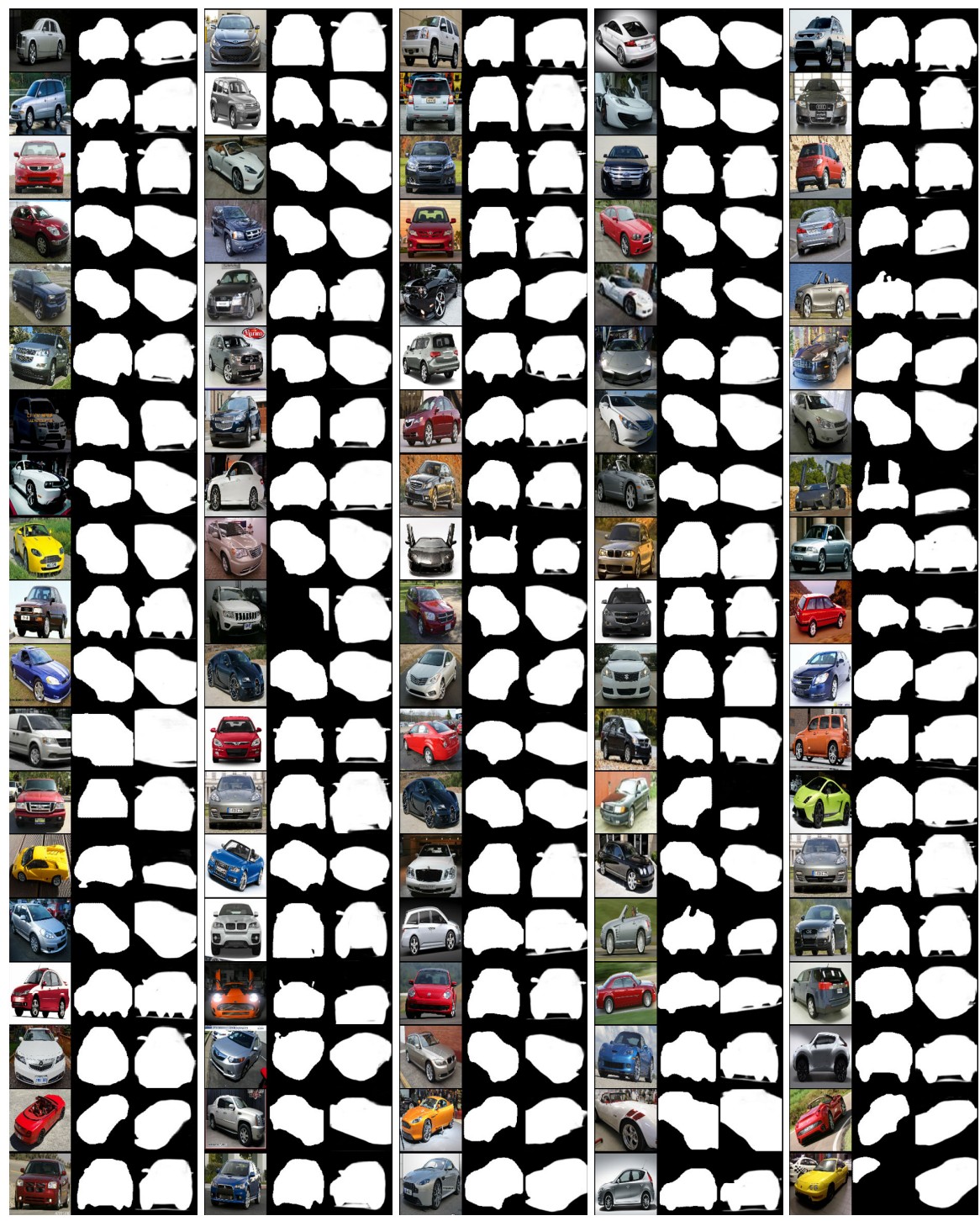

Figure 16: More randomly sampled segmentation masks on Stanford Cars.

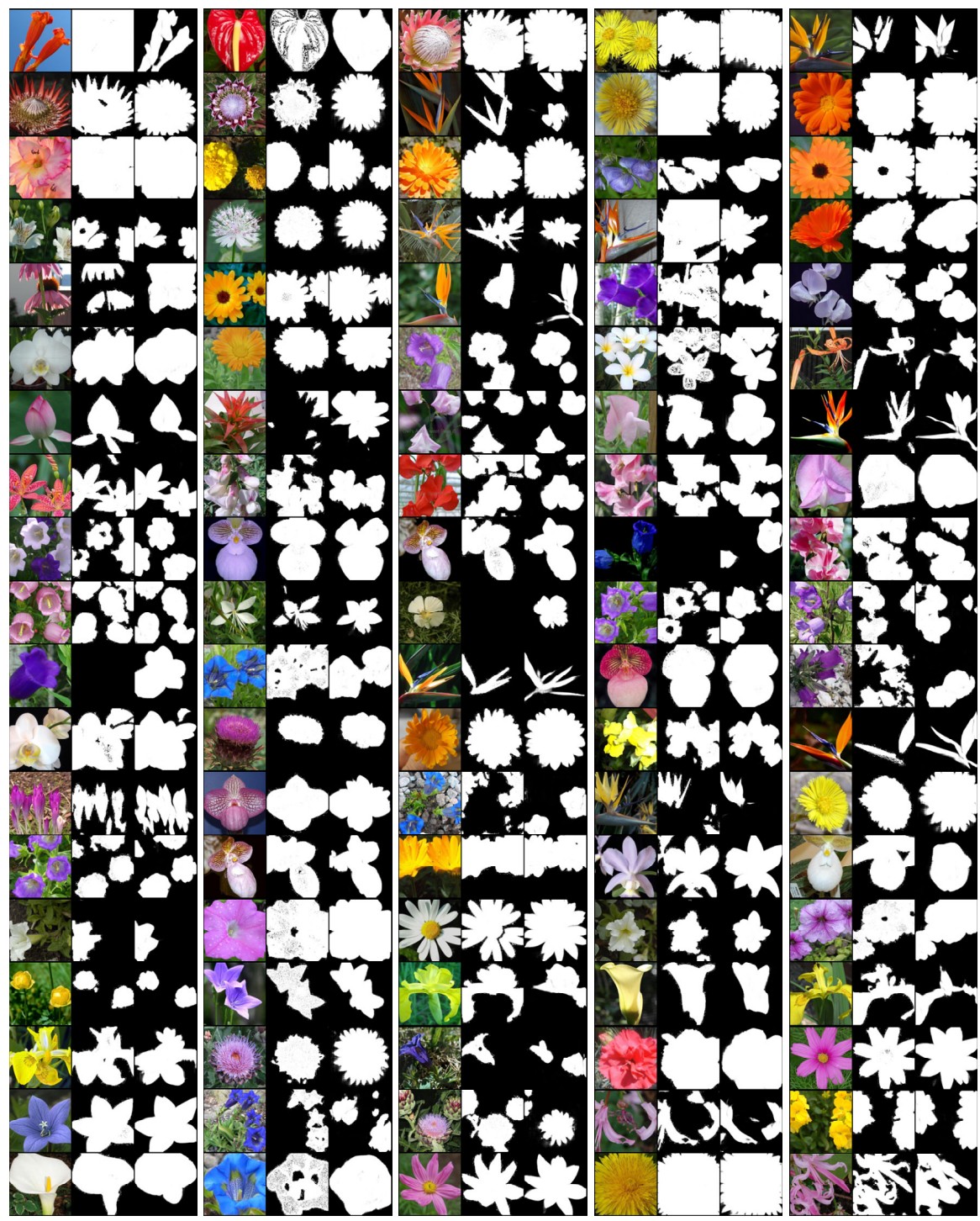

Figure 17: More randomly sampled segmentation masks on Flowers.

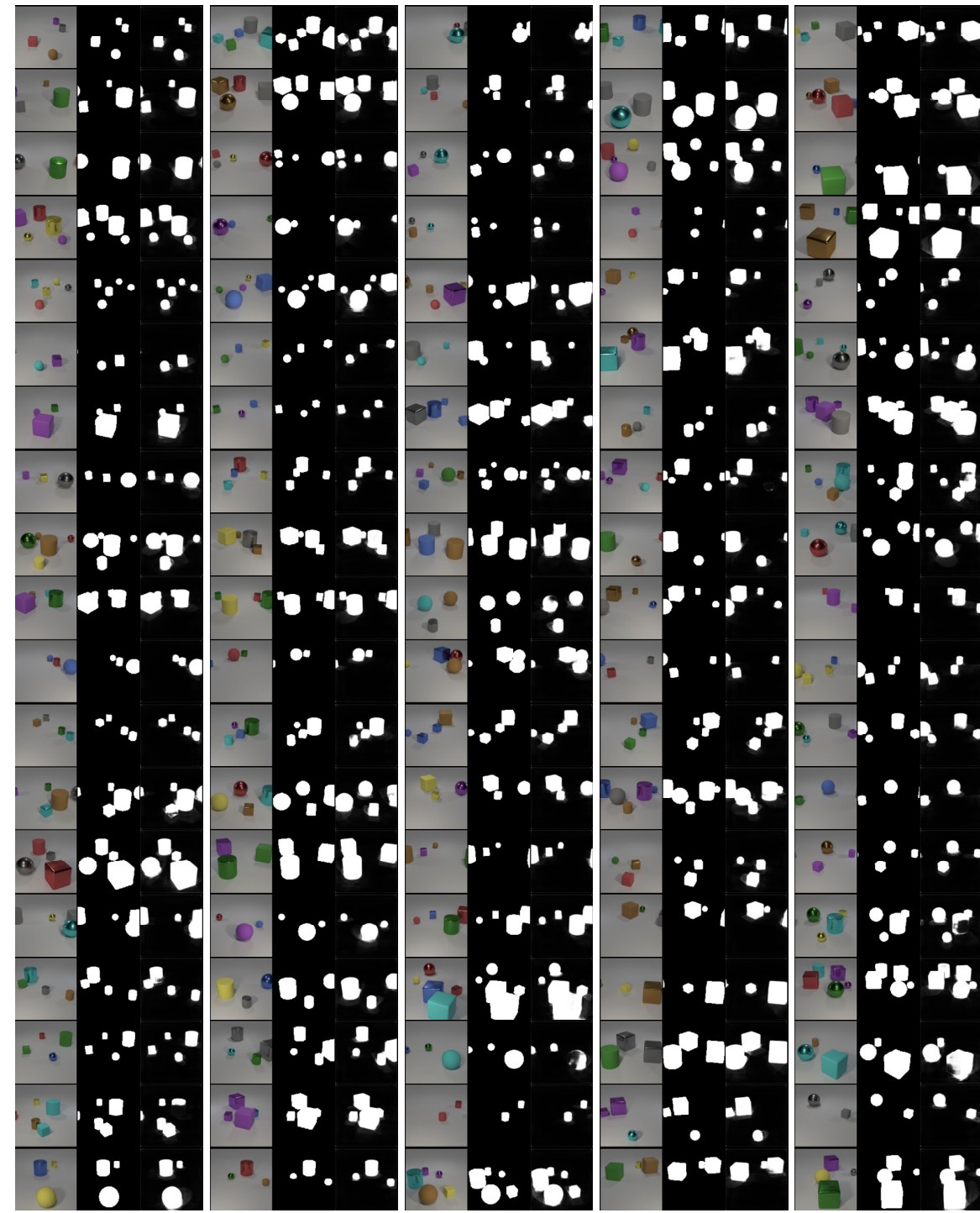

Figure 18: More randomly sampled segmentation masks on CLEVR6.