# OpenReview forum: "ComGAN: Unsupervised Disentanglement and ﻿Segmentation via Image Composition"
_NeurIPS.cc/2022/Conference — NeurIPS 2022 Accept_

### Official Review · Reviewer_rWMN · 2022-07-10

**Rating:** 6
**Confidence:** 3
**Soundness:** 3 good
**Presentation:** 2 fair
**Contribution:** 3 good

**Summary:**

This work analyses the reason for trivial solutions during mask learning in image composition GANs, and introduce a new model architecture ComGAN to solve the trivial solution issue.  Furthermore, an unsupervised object segmentation module is also involved to construct the DS-ComGAN model. DS-ComGAN can perform both disentangled image generation and object segmentation, and outperforms semi-supervised and weakly supervised baselines.

**Questions:**

The presentation of the method part is not clear enough. Please check the questions in the weaknesses part.

**Limitations:**

The limitations and potential negative societal impact have been well described.

**Strengths And Weaknesses:**

Strengths

- It is claimed that this work is the first to solve the trivial solution in disentangled image generation by changing the network architecture. The change is simple to apply and obtained significant improvement. I believe this technic can also be useful for other models and tasks.

- Both disentangled image generation and object segmentation tasks can be performed in a single framework. More importantly, the learning can be achieved in an unsupervised way by a carefully designed adversarial learning strategy.

- Sufficient experiments and analyses have been done to demonstrate the effectiveness of the proposed method.

Weaknesses

My main concerns are about the description.

- The description in sub-section 3.1 is not that readable. I suggest improving the description with a more intuitive illustration to point out the key contribution: how to avoid vanishing gradient from the network architecture perspective. Besides, some symbols seem not consistent. Do the variable f in equation 9 and the variable F in Figure 3 represent the same thing? What does the variable mean?

- The description of the mask distribution alignment could also be improved. If I understand correctly, the proposed Segmentation Networks S does not need any paired/unpaired segmentation data. For $D_m$, $\bar{x}_m$ is regarded as the real and $\hat{x}_m$ and $x_m$ are regarded as the fake.

- The learning process is not clear. Does DS-ComGAN need to be trained in two stages?  Besides, the overall objective is missing and the loss term βLbinary is introduced in the experiment section instead of the method section.

---

> ### Author Response · Authors · 2022-08-02
> **Thank you for your constructive comments**
>
> We sincerely thank you for your kind words and thoughtful comments. Below, we provide point-to-point responses to address the questions.
>
> **Comment 1**: The presentation of the method part is not clear enough. Please check.
>
> **Response 1**: Thank you for your comment. The main revisions of the method part can be summarized as follows. First, we have significantly revised sub-section 3.1 (see response 2) to improve readability. Second, we added the step of adding global features and the reason for generating $x_z$ in sub-section 3.2. Third, we completed the description of the mask distribution alignment, which includes the specific operations and effects of the alignment in sub-section 3.3.
>
> **Comment 2**:  The description in sub-section 3.1 is not that readable. [...]
>
> **Response 2**: Thank you for your suggestion. To improve the readability of sub-section 3.1, the revisions are detailed as follows. First, Remark 1 is added to show that the proposed ComGAN is a generic method and alleviates the shortcomings of the two typical methods. Second, we remove Proposition 1 and present Theorem 1 with more theoretical details. Theorem 1 proves that if the synthetic image satisfies Equation (2) and a set of constraints is satisfied, then there exists a lower bound on the gradient norm of the mask generation process. Third, we perform a simple discussion, i.e., the model avoids the first trivial solutions if the lower bound > 0. Fourth, we present Corollary 1, which maps the lower bound to a restriction on the modules in ComGAN. Eventually, the synthetic image satisfies Equation (2) and the modules meet the restrictions in Corollary 1, then the model can effectively avoid the trivial solution from the network architecture perspective.
>
> Equation (2) represents the image synthesized by ComGAN. For your convenience, it is quoted as follows,
> 	$$\bar{x}={F}(\Phi(z)) \odot {M}(\Phi(z))+{B}(\Phi(z)) \odot(1-{M}(\Phi(z)))$$.
> We tried to describe our contribution briefly with illustrations, but due to the page limit (within 9 pages), we finally chose to describe the contribution with theoretical details.
> For more details about how to avoid vanishing gradients from the perspective of network structure, please see response 1 in the official review by reviewer 19KW. or 3.1 sub-section in the revised paper.
> Thank you for your careful reading. This is a typo,  we have corrected all the typos and carefully examined the whole manuscript. The variable $f$ in Equation (9) and the variable $F$ in Figure 3 represent the same thing, i.e., the foreground variable. DS-ComGAN maximizes the mutual information between foreground variable $f$ and foreground image, so that the controllable image generation is achieved (Please see Figure 6 in the revised paper).
>
> **Comment 3**: The description of the mask distribution alignment could also be improved. [...]
>
> **Response 3**: Thank you for your comment. Yes, you understand correctly that the segmentation network does not require any segmented data input to the network. We add additional content to enhance the description of the mask distribution alignment in the revised paper.
>
> **Comment 4**: The learning process is not clear. Does DS-ComGAN need to be trained in two stages? Besides, the overall objective is missing and the loss term $\beta L_{\text {binary }}$ is introduced in the experiment section instead of the method section.
>
> **Response 4**: Thank you for your comment. We present the learning objectives for ComGAN and DS-ComGAN respectively.
>
> - ComGAN is trained by adversarial loss and binary regularization, and the learning objective is as follows,
>
>  $$ L_{\text {all}}= 	\min _{\Phi, \mathcal{F}, \mathcal{B}, \mathcal{M}} \max _{D} \mathcal{L} _D^{a d v} + \min _{\Phi, \mathcal{F}, \mathcal{B}, \mathcal{M}}\beta \mathcal{L} _{\text {binary}}$$
>
> - DS-ComGAN performs two unsupervised tasks, image disentanglement and object segmentation.
>
> $$ L_{\text {all}} = \max _{D_f, D_b} L _{\text {info}} + \min _{G, \mathcal{F}, \mathcal{B}, \mathcal{M}} \max _{D_z} \mathcal{L} _{D_z}^{a d v} + \min _{S} \max _{D_m} \mathcal{L} _{D_m}^{a d v} +   \min _{S} \lambda\mathcal{L} _{\text {cons}}$$
>
> The disentangled image generation task is performed in a single stage, and its overall loss function is $\max _{D_f, D_b} L _{\text {info}} + \min _{G, \mathcal{F}, \mathcal{B}, \mathcal{M}} \max _{D_z} \mathcal{L} _{D_z}^{a d v} $.  The segmentation network needs the images and the semantic mask generated by the disentanglement module. Therefore, the segmentation task needs to be trained in two stages and its overall loss function is $+ \min _{G, \mathcal{F}, \mathcal{B}, \mathcal{M}} \max _{D_z} \mathcal{L} _{D_z}^{a d v} + \min _{S} \max _{D_m} \mathcal{L} _{D_m}^{a d v} +   \min _{S} \lambda\mathcal{L} _{\text {cons}}$.
>
> Unfortunately, due to page limits, we could not place it in the methods section, but rather B.2 and B.3 in the revised supporting material.

---

> > ### Comment · Reviewer_rWMN · 2022-08-08
> > **Response to Author Feedback**
> >
> > Thanks for the authors’ effort in the feedback. My concerns have been addressed. I appreciate the additional detailed theoretical analysis on the problem.

---

> > > ### Author Response · Authors · 2022-08-09
> > > **Many thanks for the response**
> > >
> > > Thank you for your response and appreciation. We are glad that our feedback addressed your concerns. We will be encouraged if you would like to raise your rating accordingly. Please let us know if there is anything else we can do to make the paper better.

---

> ### Author Response · Authors · 2022-08-07
> **Looking forward to your feedback**
>
> Dear Reviewer  rWMN,
>
> Thanks again for your valued advice! We have responded to your initial comments. We are looking forward to your feedback and will be happy to answer any further questions you may have.
>
> Thank you, author

---

### Official Review · Reviewer_19KW · 2022-07-11

**Rating:** 6
**Confidence:** 4
**Soundness:** 3 good
**Presentation:** 3 good
**Contribution:** 3 good

**Summary:**

The authors point out two factors that lead a scene decomposition model to fall into trivial solutions in a mathematical analysis.
Those are related to the vanishing gradient phenomena that goes into a mask generator.
To avoid these, they propose a novel network architecture, where features for generating decomposed scene elements are composed of the ones that used for generating the entire scene at once.
With this architecture, they achieved the SOTA scores on both the mask prediction and the image quality evaluation metrics.

**Questions:**

Questions
- Why the proposed network is helpful for avoiding the vanishing gradient issue?
- Why the performance of PerturbGAN was not compared?

Suggestions
- Maybe simplegan —> finegan at the line 221?

**Limitations:**

Authors did not address the limitations and potential negative societal impact of their work.

Suggestions
- I want to know the authors’ opinion on how much would it be difficult to apply this method on coarse grained datasets like ImageNet?

**Strengths And Weaknesses:**

Strengths
- The authors tackle the important problem in the scene component generation models.
- They propose a novel architecture for robust mask generation based on the theoretical analysis on the problem.
- The authors did a thorough ablation study to show that each element proposed are all effective.
- The comparison of both quantitative and qualitative results with previous works imply that the proposed method is effective in boosting the scene decomposition performance.

Weaknesses
- The connection between the theoretical analysis and the proposed architecture design is not well established. More details are needed to understand why the proposed architecture is helpful for model to not fall into the vanishing gradient phenomena.

---

> ### Author Response · Authors · 2022-08-02
> **Thank you for your constructive comments**
>
> We sincerely thank you for your time and constructive comments. Below, we provide point-to-point replies to your comments in order and hopefully resolve the remaining questions you have.
>
> **Comment 1:** Why the proposed network is helpful for avoiding the vanishing gradient issue?
>
> **Response 1:** Thank you for your comment.  We summarize the following three points to explain this issue.
> -  Our model is a generic image compositional generation method.
>
> We review that the image synthesized by ComGAN can be written as follows:
>  $$\bar x = {F}(\Phi(z))\odot{M}(\Phi(z))+{B}(\Phi(z))\odot(1-{M}(\Phi(z)))$$
>
> $\mathbf{Rmark 1}$:
> *This form generalizes two typical image compositional generation methods:*
>
>  *-  If only $\Phi(\cdot)$ in ${B}$ is an identity map, i.e., ${B}(\Phi(z))={B}(z)$, then this form is equivalent to model $\Pi_{1}$, that is, two independent generators synthesize a composite image where shared features exist in the foreground and mask generation. To our knowledge, FineGAN, MixNMatch, OneGAN, C3-GAN and Labels4Free, etc. can be written as model $\Pi_{1}$.*
>
> *- If $\Phi(\cdot)$ is an identity map, i.e., $\Phi(z)=z$, then this form is equivalent to model $\Pi_{2}$, that is, three independent generators synthesize a composite image where foreground, background and mask are generated by three generators respectively. To our knowledge, PerturbGAN and CGN, etc. can be written as model $\Pi_{2}$.*
>
> From the above observations, we notice that both the two typical methods have a common shortcoming.
> Both models $\Pi_{1}$ and $\Pi_{2}$ contain an independent background generation process, which not only ignores the feature connection between foreground, background, and mask, but also may be limited by shortcomings of GANs, such as mode collapse.
>
> - Our model presents a lower bound on the gradient 2-norm.
>
> $\mathbf{Theorem 1}$: Given a generation $G_{\theta}$ composed of a decoder $\Phi_{\theta_\phi}: \mathcal{Z} \rightarrow \phi$  and three subnetworks $F_{\theta_f}, B_{\theta_b}$ and $M_{\theta_m}$: $\phi \rightarrow\mathcal{X}$. Let $D$ be a discriminator and $D^*(G^*(\cdot))$ be Nash equilibrium. If the generated images satisfy that $\bar x = {F}(\Phi(z))\odot{M}(\Phi(z))+{B}(\Phi(z))\odot(1-{M}(\Phi(z)))$, $\lVert D(G(\cdot))- D^*(G^*(\cdot))  \rVert <\epsilon$, $\max$ { $\mathbb{E} _{ \phi\sim p(\Phi(z))} \rVert J _{\theta_f} F ( \phi )  \rVert$, $\mathbb{E} _{\phi \sim p(\Phi(z))} \rVert J _{\theta_b} {B} ( \phi )  \rVert $} $\leq \delta ^{2}$ and $\lVert L^{adv} _D \rVert \geq \sigma$, then
>
> $$\rVert \nabla_{(\theta_{\phi}, \theta_m)} \mathbb{E} _{z \sim p(z)} \rVert \log(1-D(F(\Phi(z))))\rVert^2_2 \geq \sigma^2 -\delta \frac{\epsilon^2}{1/2 - \epsilon^2}$$
>
> *Proof: Please see Theorem 1 in the revised paper.*
>
> We denote that $\rho=\sigma^2 -  \delta^{2} \frac{\epsilon^{2}}{(1/2-\epsilon)^{2}}$ and trivial masks as
>  $\bar x^*_m  ={M} _{\theta^*_m} (\Phi _{\theta ^*_\phi}(z))$. If $\rho>0$, which implies that the $\Phi$ and ${M}$ are updated. The updated masks are  $\bar x ^+ _m  ={M} _{\theta ^+ _m} ( \Phi _{\theta ^+_\phi}(z))$, which means that the model can escape from the first trivial solutions, i.e. $ \bar{x}^+_m \neq \bar x^*_m$ and $\partial L ^{adv} _D/ \partial \bar{x}^+_m\neq0$.
>
> - Module restrictions in our model.
>
> $\mathbf{Corollary 1}$: The following modules restrictions help the model avoid trivial solutions: the ${F}$ and ${B}$ are lightweight and differential,  the ${M}$ is a shallow network and the capacity of $\Phi$ is enough.
>
> *Proof: By Theorem 1, the larger the value of $\rho$, the easier it is for the model to escape from the trivial solution. We observe that $\max$ { $\mathbb{E} _{ \phi \sim p( \Phi ( z ) ) } \rVert J _{\theta_f} F ( \phi )  \rVert$, $\mathbb{E} _{\phi \sim p(\Phi(z))} \rVert J _{\theta_b} {B} ( \phi )  \rVert $} $\leq \delta ^{2}$, which means that reducing the parameter of $\theta_f$ and $\theta_b$ can effectively increase $\rho$. Notice that when $\theta_f$ and $\theta_b$ have too few parameters, or even no parameters, our model degrades into a raw GAN. Therefore, $F$ and $B$ should be designed as lightweight modules (e.g., adding residual connections), so as to enhance the capacity of the module with fewer parameters. Furthermore, the $\rho$ is actually the gradient norm from the decoder $\Phi$. It is a natural idea to increase $\rho$ by raising the capacity of $\Phi$.
> In addition,  the update of $\Phi$ does not necessarily mean the update of $M$. If $M$ is a deep neural network, it may not be able to map the fluctuations of features to the mask space, i.e. ${M}(\phi) \approx {M}(\phi +\triangle\phi)$. Therefore, $M$ should be designed as a shallow neural network.  As for the second trivial solution, $\bar x_f = \bar x_b$ is a fragile equilibrium. We can break the identical mapping, i.e. ${F} _{\theta_f}(\phi) = {B} _{\theta_b}(\phi)$,  by changing  parameters of $\theta_f$ and $\theta_b$ or modifying the structure of ${F}$ and ${B}$.*

---

> > ### Author Response · Authors · 2022-08-02
> > **Thank you for your constructive comments (Part 2)**
> >
> > **Comment 2**:  The connection between the theoretical analysis and  [...].
> >
> > **Response 2**: Thank you for your comment. We give the connection between the proposed model and the theoretical analysis in Response 1. In addition, we give a visualization to show that the proposed architecture helps the model not to fall into trivial solutions. In the revised paper, we trace the gradient norm of Model $\Pi_1$, Model $\Pi_2$ and our model and plot it in Figure 5.
> > It is clear from Figure 5 that the gradients norm of mask networks in model $\Pi_1$ and model $\Pi_2$  converge to zero and the model degrades to a raw GAN, while our method effectively avoids vanishing gradients.
> >
> > **Comment 3**:  Why the performance of PerturbGAN was not compared?
> >
> > **Response 3**: Thank you for your comment. As reported by IEM+SegNet, IEM+SegNet outperforms PerturbGAN by a big margin, while the performance of our method outperforms IEM+SegNet. Therefore, we do not compare our method with PerturbGAN. The suggestion improves the persuasiveness of this paper, in the revised paper we compare the performance of PertureGAN and add it to Table 5.
> >
> > **Comment 4**:  Maybe simplegan --> finegan at the line 221?
> >
> > **Response 4**: Thank you for your comment. Our baseline model is SimpleGAN and not FineGAN.  Notice that SimpleGAN is also the baseline model and backbone of FineGAN. For a fair comparison, we build the baseline model based on SimpleGAN, which generates global features as a feature decoder.
> >
> > **Comment 5**:  Authors did not address the limitations and potential negative societal impact of their work.
> >
> > **Response 5**: Thank you for your comment.  We describe the limitations of our work in the Conclusion, which is quoted as follows:
> >
> > "As the limitation of method, DS-ComGAN has struggled to achieve the desired performance when the foreground object features are highly diverse (e.g., HKU-IS )."
> >
> > To clarify the potential negative societal impact of our work, some comments have been added to the Conclusion of the revised manuscript. For your convenience, it is quoted as follows:
> >
> > "DS-ComGAN performs excellently in controlled image synthesis tasks, which may cause the incidence of image falsification."
> >
> > **Comment 6**:  I want to know the authors' opinion on how much would it be difficult to apply this method on coarse grained datasets like ImageNet?
> >
> > **Response 6**:  Thanks for this good question!
> > Historically, the existing methods that perform well on fine-grained datasets have been difficult to migrate to coarse-grained datasets. The issue is not only related to the model capacity, but also to both the dataset-sensitive regularization and hyperparameters. It is notable that our method contains a few hyperparameters and it is robust to these hyperparameters. Hence along the lines of the issue, we migrated DS-ComGAN directly to the coarse-grained dataset CIFAR-10 for experiments (Compared with ImageNet, we trained DS-ComGAN on CIFAIR-10 within 24 hours).
> > We set that dimensionality of latents $N=10$, binary regularization weight $\beta=0.5$ and scale all the images to 64 $\times$ 64 pixels. Although the model has not exhibited superior performance as on the fine-grained dataset, the model shows similar results on CIFAIR-10 without other fine-tuning, that is, DS-ComGAN synthesizes the images and the corresponding semantic masks. More visualized experimental results are added to the supplementary material ( Please see D.3 in the revised supplementary materials ). Experiments on CIFAIR-10 once again demonstrated that our method is flexible and robust to different datasets. Returning to the question, we consider that there are at least two difficulties in applying our method to the ImageNet dataset. First, the capacity of the model should be improved to learn more features. Second, since ImageNet has a large number of categories, the model needs additional regularization or supervised information to learn the significant category variations.

---

> > > ### Comment · Reviewer_19KW · 2022-08-08
> > > **Response to the authors' rebuttal**
> > >
> > > Dear Authors,
> > >
> > > Thank you for the additional work!
> > > Your response did resolve most of my concerns, so I adjusted my initial evaluation.
> > > The section 3.1 in the revised version seems more convincing than the prior one.
> > > I also like the additional empirical results regarding the gradient values observed during a training stage as given in Figure 5!

---

> ### Author Response · Authors · 2022-08-07
> **Looking forward to your feedback**
>
> Dear Reviewer 19KW,
>
> Thanks again for your valued advice! We have responded to your initial comments. We are looking forward to your feedback and will be happy to answer any further questions you may have.
>
> Thank you, author

---

### Official Review · Reviewer_Tkuw · 2022-07-11

**Rating:** 6
**Confidence:** 3
**Soundness:** 3 good
**Presentation:** 2 fair
**Contribution:** 3 good

**Summary:**

The paper considers the task of learning GANs that decompose the image formation process into foreground, background and mask generation and composition. Compared to previous methods, the proposed ComGAN aims to avoid trivial solutions (where masks do not correspond to foreground objects) mainly through the network architecture instead of regularizations, which often require extensive hyperparameter searches for suitable regularization strengths.

**Questions:**

- l. 47-60 could be moved to the related work section
- l. 294 refers to the wrong figure
- l. 298: how does fine-tuning mitigate masks that are inconsistent with foreground images? Also, what is the effect of the mask distribution alignment (l. 206)? Why not train the segmentation model S with Eq. (13) only?
- It would be helpful to use the same evaluation protocol as in FineGAN (based on 30k samples) for Tab. 3. The reported Inception Scores seem very low.
- See also weaknesses for questions

**Limitations:**

Limitations and potential negative societal impact have been addressed adequately.

**Strengths And Weaknesses:**

- Strengths
    - The proposed approach simplifies the design of compositional GANs compared to previous methods and demonstrates improved performance in terms of synthesis quality as well as unsupervised segmentation performance.
    - Compared to similar compositional generative models like FineGAN [25], the supervision requirements regarding weak background supervision is further reduced.
    - Finding a compositional generator architecture that is stable to train has many applications beyond GANs. Thus, the work is potentially interesting for a larger audience.
    - Although there are still a few hyperparameters (mask consistency loss weight, binary regularization weight, dimensionality of latents, relative sizes of subnetworks), experiments demonstrate some robustness to these parameters.
- Weaknesses
    - The core idea and differences to previous approaches are not clearly stated.
    - The requirements for Proposition 1 regarding what it means for an architecture to be "similar to ComGAN" are not stated clearly. I assume the key point is that M consists of only an sigmoid layer. However, the formulation in l. 149 seems to be the only place where this is stated and even there it remains vague and could be interpreted as containing the same layers as F and B and in addition a sigmoid layer. The latter interpretation is also what Fig. 3 suggests (albeit with one residual block less). If this (a shared decoder with a minimal mask decoder) is the key idea of the paper, it should be communicated more clearly and probably also on a higher level already in the introduction. Without clear restrictions on the architecture, one could also think that FineGAN satisfies the requirements with G set to the identity.
    - The lemmas, propositions and proofs seem a bit vague as they do not clearly state assumptions or define all involved quantities. In l. 160-161 it is not clear what is meant by $\bar{x}_m = M^{-1}(G(z))$ - why would the input of M equal its output? Intuitively, I also don't see how "it is clear that any change of foreground or background affects the mask, [...]". Since F and B do contain additional layers, foreground and background could be affected by changes in those layers even though G(z) and hence the mask would remain unaffected, no? The other way, that any change of the mask affects both foreground and background, seems to be true.
    - Motivation for DS-ComGAN architecture is unclear: Why is the $\bar{x}_z$ output needed in addition to the output composited from foreground, background and mask?

---

> ### Author Response · Authors · 2022-08-02
> **Thank you for your constructive comments**
>
> We sincerely thank you for your time and constructive comments. We hope our detailed responses below would resolve the remaining questions you have.
>
> **Comment 1**: l. 47-60 could be moved to the related work section.
>
> **Response 1**:  Thank you for your suggestion. We move l. 47-60 to the related work section and rewrite the introduction and related work section in the revised manuscript.  In the introduction, we describe the negative role of trivial solutions on two related tasks and highlight the differences between DS-ComGAN and previous methods on both tasks.
>
> **Comment 2**: l. 294 refers to the wrong figure.
>
> **Response 2**: Thank you for the careful reading. We have corrected the index and double-checked the entire manuscript.
>
> **Comment 3**: l. 298: how does fine-tuning mitigate masks that are inconsistent with foreground images? [...]
>
> **Response 3**: Thank you for your comment. We think that this inconsistency is caused by the adversarial training strategy, which is similar to a model collapse. However, we alleviate the inconsistency issue by increasing the weight of $L_{cons}$, i.e., $\lambda$  to force the segmentation network to learn more mask features. The main effect of mask distribution alignment is that the model learns the low dimensional manifold of the masks, so that the predicted segmentation masks contain more details and are clearer.
> Please see Fig. 8 in the revised paper, the predicted masks segment visual details precisely, such as the legs of birds and the rearview mirrors of cars. For more segmentation results, please see D.4 in the revised supporting material. The $L_{cons}$ in  Eq. (13) is a pixel-wise loss. This kind of loss reduces the correlation between pixels. If we train the segmentation model $S$ with the $L_{cons}$ only, the predicted segmentation masks may be blurred and lack details.
>
> **Comment 4**: It would be helpful to use the same evaluation protocol as in FineGAN (based on 30k samples) for Tab. 3. The reported Inception Scores seem very low.
>
> **Response 4**: Thank you for your suggestion. We adopt the same evaluation protocol as SSG-GAN, instead of FineGAN. According to the official code of SSG-GAN, they generate 20K samples to evaluate the model performance. The benefits of choosing SSG-GAN as our evaluation protocol are as follows: 1) SSG-GAN compares a wide range of GAN-based generation models, including weakly supervised models such as FineGAN and semi-supervised models such as Triangle-GAN. 2) SSG-GAN is the SOTA semi-supervised model in the image disentanglement generation task.  Therefore, it is clear from Table 3 that our unsupervised method outperforms the SSG-GAN, which highlights the advantages of our model, i.e., our model is unsupervised and outperforms SOTA. The reported Inception Scores is taken from Table 2 in SSG-GAN. Moreover, according to the official code of FineGAN, they use the fine-tuned version for computing the Inception score, where the inception model is fine-tuned on all the 200 categories (for birds). This may be the reason why the reported Inception Scores seem to be low.
>
> **Comment 5**: The core idea and differences to previous approaches are not clearly stated.
>
> **Response 5**: Thank you for your comment. Our core idea can be summarized in a sentence, i.e., "find the causes of trivial solutions, solve trivial solutions via the perspective of architecture and generalize the model to downstream tasks."   One of our main contributions is to focus on solving trivial solutions from the perspective of architecture. The key points of this method are as follows: 1) the model is a generic image compositional generation method. 2) the model presents a lower bound on the gradient 2-norm of the mask generation process. 3) the module restrictions help the model to better avoid trivial solutions. For more details,  please see response 1 in the official review by reviewer 19KW.
>
> The differences with the previous methods are summarized into four aspects, which are 1) differences in alleviating trivial solutions; 2) differences in generative models; 3) differences in image disentanglement methods; 4) differences in unsupervised segmentation methods.
>
> - Differences in alleviating trivial solutions.
>
> To the best of our knowledge, no previous work has indicated that the source of trivial solutions is vanishing gradients on the mask. Our method is also the first to solve trivial solutions from the perspective of architecture.  Existing work alleviates trivial solutions in two ways. One way is to add supervised information, such as CGN avoids trivial solutions by adding pre-trained U2-Net. Another is to design clever regularization and fine-tune the parameters.
>
> - Differences in generative models.
>
> Please see response 1 in the official review by reviewer 19KW.

---

> > ### Author Response · Authors · 2022-08-02
> > **Thank you for your constructive comments (Part 2)**
> >
> > - Differences in image disentanglement methods.
> >
> > Existing methods rely on additional supervised information to learn the distinction of image regions. Notice that ComGAN achieves foreground-background disentanglement in an unsupervised way. As a result, we add more global information to the shared features in ComGAN and maximize the mutual information between variables and images.  Our image disentanglement method is unsupervised and simplifies the previous hierarchical generative network and outperforms the SOTA semi-supervised and weakly supervised image disentanglement methods.
> >
> > - Differences in unsupervised segmentation methods.
> >
> > Existing methods rely on strong assumptions, such as that the foreground and background are largely independent, which limits their applicability. Different from these methods, we train a segmentation network using the images and semantic masks synthesized by the ComGAN variants. By adversarial training strategy, the image distribution and the mask distribution are aligned. Furthermore, a consistency regularization is introduced to ensure that the predicted masks are consistent with the input images. Our unsupervised segmentation method relies only on mild assumption 1 and outperforms the SOTA unsupervised segmentation methods.
> >
> > *All four of these aspects are added to the revised paper, so as to highlight the differences between our methods and the previous methods.*
> >
> > **Comment 6**: The requirements for Proposition 1 regarding what it means for an architecture to be "similar to ComGAN" are not stated clearly, [...]
> >
> > **Response 6**:  Thank you for your suggestion. To clearly demonstrate that the proposed architecture can help the model avoid trivial solutions,  we remove Proposition 1 and present  Remark 1, Theorem 1 and Corollary 1 in the revised paper.
> > In the revised paper, we have replaced "the generated images satisfy that $\bar x = {F}(\Phi(z))\odot{M}(\Phi(z))+{B}(\Phi(z))\odot(1-{M}(\Phi(z)))$" with "similar to ComGAN" for clarity.
> > As stated in Corollary 1, a minimal ${M}$ can help the model avoid trivial solutions. Because if the ${M}$ is a deep neural network, it may fail to map the fluctuations of shared features to the mask, i.e., ${M}(\phi) \approx {M}(\phi +\triangle\phi)$. If we fix the ${M}$ as a sigmoid layer, although the model can avoid trivial solutions, the generalization of the model decreases dramatically. Therefore, we perform the experiment of relaxing constraints on the ${M}$ (Please see C.3 in the revised support material). We improve the capacity of  ${M}$ by adding residual blocks. The experimental results show that the model is robust to the capacity of  ${M}$, and the ${M}$ with one residual block improves the model performance.
> >
> > We place restrictions on each module in Corollary 1. For your convenience, it is quoted as follows:
> >
> > $\mathbf{Corollary1}$: The following modules restrictions help the model avoid trivial solutions: the ${F}$ and ${B}$ are lightweight and differential, the ${M}$ is a shallow network and the capacity of $\Phi$ is enough.
> >
> > FineGAN can be written as model $\Pi_{1}$ in Remark 1. Furthermore, in the revised paper experiments, we plot the gradient norm figure, where the model $\Pi_{1}$ is taken from the official code of FineGAN. From the Figure 5, we intuitively observe that the gradients of both the mask and foreground network of model $\Pi_{1}$ converge to 0.
> >
> > **Comment 7**: The lemmas, propositions and proofs seem a bit vague as they do not clearly state assumptions or define all involved quantities, [...]
> >
> > **Response 7**: Thank you for your comment. This is a typo in I. 160-161,  we have corrected all the typos and carefully examined the whole manuscript.  Your inference is correct when the  $F$ and $B$ contain a large number of parameters and layers. However, we get a lower bound in Theorem 1 that is related to the parameters of $F$ and $B$.  One way to improve this lower bound is to reduce the parameters of $F$ and $B$. Thus, it is possible that this statement is correct only in Corollary 1, i.e., "Any change in the foreground or background affects the mask. "
> >
> > **Comment 8**: Motivation for DS-ComGAN architecture is unclear: Why is the $x_z$ output needed in addition to the output composited from foreground, background and mask?
> >
> > **Response 8**: Thank you for your comment. The role of $x_z$ is to add global information to the features of $G$ via adversarial training.  Since the features in $G$ are fed into three subnetworks, if the features contain much local information, the model focuses on the abrupt regions. The operation is necessary for specific datasets such as Stanford Cars, because it permits the mask to capture the across-the-board distinction between the foreground and background. Please see D.2 in the revised supporting material for the ablation study on global feature extraction.

---

> > > ### Comment · Reviewer_Tkuw · 2022-08-08
> > > **Thanks**
> > >
> > > Thank you for your comments. My main concerns regarding the evaluation and the clarity of the method were addressed. I think Sec. 3.1 is much more readable now and Remark 1 is very helpful to understand the proposed changes in relation to existing works (minor suggestion: maybe include the examples for priors works that use $\Pi_1$, $\Pi_2$ that you gave in the answer to another reviewer, too). I raised my rating to 6.

---

> ### Author Response · Authors · 2022-08-07
> **Looking forward to your feedback**
>
> Dear Reviewer Tkuw,
>
> Thanks again for your valued advice! We have responded to your initial comments. We are looking forward to your feedback and will be happy to answer any further questions you may have.
>
> Thank you, author

---

### Meta-Review · Area_Chair_kXKd · 2022-08-26

**Recommendation:** Accept
**Confidence:** Certain

**Metareview:**

The paper proposes a compositional GAN model with a novel network architecture that solves the vanishing gradient problem underlying trivial solutions. The proposed model achieves strong results on image disentanglement and unsupervised segmentation tasks.

The rebuttals by the authors have successfully addressed most of the concerns of the reviewers. All the reviewers are positive about this paper. Reviewer Tkuw's main concerns regarding the evaluation and the clarity of the method were addressed. The reviewer raised the rating. Reviewer 19KW felt positive about section 3.1 in the revised version and the additional empirical results regarding the gradient values observed during a training stage as given in Figure 5. The reviewer also updated the initial rating. Reviewer rWMN's concerns have also been addressed. The reviewer appreciates the additional detailed theoretical analysis on the problem.

**Award:**

No

---

### Decision · Program_Chairs · 2022-09-14

Accept